# Hydrometeor classification of quasi-vertical profiles of polarimetric radar measurements using a top-down iterative hierarchical clustering method

Maryna Lukach[1,2], David Dufton[1,2], Jonathan Crosier[3,4], Joshua M. Hampton[1,2], Lindsay Bennett[1,2] and Ryan R. Neely III[1,2]

[1]National Centre for Atmospheric Sciences, Leeds, United Kingdom

[2] School of Earth and Environment, University of Leeds, Leeds, United Kingdom

[3]National Centre for Atmospheric Sciences, University of Manchester, United Kingdom

[4]Department of Earth and Environmental Sciences, University of Manchester, Manchester, United Kingdom

*Correspondence to*: Maryna Lukach (maryna.lukach@ncas.ac.uk)

**Abstract.**

Correct, timely and meaningful interpretation of polarimetric weather radar observations requires an accurate understanding of hydrometeors and their associated microphysical processes along with well-developed techniques that automatize their recognition in both the spatial and temporal dimensions of the data. This study presents a novel technique for identifying different types of hydrometeors from Quasi-Vertical Profiles (QVP). In this new technique, the hydrometeor types are identified as clusters belonging to a hierarchical structure. The number of different hydrometeor types in the data is not predefined and the method obtains the optimal number of clusters through a recursive process. The optimal clustering is then used to label the original data. Initial results using observations from the NCAS X-band dual-polarization Doppler weather radar (NXPol) show that the technique provides stable and consistent results. Comparison with available airborne in situ measurements also indicates the value of this novel method for providing a physical delineation of radar observations. Although this demonstration uses NXPol data, the technique is generally applicable to similar multivariate data from other radar observations.

# 1 Introduction

The task of radar-based hydrometeor classification (HC) can be broadly defined as the recognition of different hydrometeor types in the atmosphere as represented by the various observed moments collected by weather radar. In general, HC is able to label radar signatures observed at any one time with physical properties and, over a period of time, the evolution of these labels can provide insight into the underlying atmospheric processes. As such HC has many impactful applications: HC simplifies the detection of the melting layer (Baldini and Gorgucci, 2006), HC is necessary for obtaining accurate estimates of precipitation quantities (Giangrande and Ryzhkov, 2008) and HC provides critical information for improving modelling of physical processes in the atmosphere (Vivekanandan et al., 1999).

Radar-based HC requires an extensive and accurate (i.e. expert) knowledge of the physical properties of both multivariate polarimetric observations and the hydrometeor particles themselves (Hall et al., 1984). Achieving an accurate and precise radar-based HC is difficult due to the deficiencies (such as low spatial-temporal resolution) and inaccuracies (such as attenuation) that are inevitable in all radar measurements. The process of HC is made even more difficult when this analysis needs to be performed during the operational processing of the radar observations where there is a lack of time for expert assessment. Therefore, automatization of spatial and temporal analysis of multivariate polarimetric data is an important task for which an advanced and well-tested technique should be developed and utilized.

The development of radar-based HC started in the 1980s and 1990s with the works of Hall et al. (1984), Hendry and Antar (1984), Aydin et al. (1986), Straka and Zrnić (1993) and Straka (1996). Further refinement and development of automatic HC algorithms included the application of fuzzy-logic (Straka et al., 2000; Liu and Chandrasekar, 2000), machine-learning techniques (such as the identification of clusters representing data-wise similarities) (Wen et al., 2015; Grazioli et al., 2015; Besic et al., 2016; and Ribaud et al., 2019) and neural networks (Wang et al., 2017).

Modern radar-based HC methods (Straka et al, 1996; Liu and Chandrasekar, 2000; Al-Sakka et al., 2013; Grazioli et al., 2015; Besic et al., 2016; Besic et al., 2018; Bechini and Chandrasekar, 2015; and Wang et al., 2017) are based on the multivariate data of polarimetric Doppler radar observations. This includes (but is not limited to): the horizontal reflectivity factor $Z_H$, differential reflectivity $Z_{DR}$, the copolar correlation coefficient $\rho_{HV}$, differential phase shift on propagation $\Phi_{DP}$, and specific differential phase $K_{DP}$ (for definitions see Bringi and Chandrasekar, 2001) as well as associated derived variables (e.g. standard deviation). Additionally, temperature and other meteorological data, retrieved from radiosondes or NWP models, are often utilized (Grazioli et al., 2015; Wen et al., 2015).

In most existing radar-based HC methods, the multivariate input data are analysed per measurement voxel and determined classes are assigned to the hydrometeor types only according to their characteristics. Such an approach neglects intra-class relationships and the temporal evolution of the identified classes. This valuable information can also be used in the labelling of the hydrometeor types and the identification of corresponding microphysical processes. Additionally, almost all methods within the existing literature are based on theoretical assumptions on the scattering properties of observed particles and/or are only applicable for a defined (i.e. previously recognized) number and type of classes. Both of these aspects of existing HC

Deleted: ,

Deleted: and neural networks (Straka et al., 2000; Liu and Chandrasekar, 2000;

Deleted: Wang et al., 2017;

Deleted: ,

Deleted: logarithmic

Deleted: at horizontal polarization

methods are limitations. For example, theoretical assumptions about the scattering properties of ice-phase hydrometeors are
very uncertain due to unknown size distributions, varying dielectric properties, fall orientation, and their diverse and often
complex geometry (Johnson et al., 2012). A pre-defined number of classes or hydrometeor types is subjective and creates
artificial boundaries for algorithms and, thus, subtle differences in undefined sub-classes are masked, which inhibits
identification of the underlying microphysical processes.
Thus, in this study we take a different approach and ask the following question: can a data-driven HC approach provide an
optimal number of classes from the observations? We define the optimal number as the lowest number of classes representing
all pronounced dissimilarities in the input data. Once the optimal set of classes is identified, the burden of analysis in this
approach is to relate the identified clusters of radar signatures to possible physical properties of hydrometeors. Thus, this
approach does not impose a predefined physical view on the observations but provides a framework for a more efficient
physical interpretation of the properties of the resulting clusters of observed multivariate data in which subtle differences and
intra-cluster relations are easier to identify. In this sense, this approach inverts the procedure of existing methods. Additionally,
we ask whether such an approach can be used to provide information on the temporal evolution of the identified hydrometeors
and reveal relationships between the identified classes. Such information is key for identifying the processes that lead to high
impact weather (i.e. flooding) and improving the physical parametrizations in NWP.
The point of this study is not to create a set of cluster characteristics that could be applied to other datasets. Rather the goal is
to demonstrate the viability of this type of data-driven methodology for creating a set of labelled clusters (i.e. hydrometeor
classes). As such, the comparison to in situ data and labelling done as part of this study is only shown as an example of how
this tool can be used.
The existing data-driven unsupervised (Grazioli et al., 2015; Ribaud et al., 2019 and Tiira and Moisseev, 2019) and semi-
supervised approaches (Bechini and Chandrasekar (2015), Besic et al., 2016; Wang et al., 2017, Roberto et al., 2017 and Besic
et al., 2018) only partially provide an answer to the first question (Grazioli et al., 2015) and do not consider the temporal
evolution or dependencies between the identified classes. The approach described here performs an unsupervised clustering of
quasi-vertical profiles (QVP). QVP were first used in Kumjian (2012) and Ryzhkov et al. (2016) as a way of constructing a
substitute for a vertical profile from a scan conducted at constant elevation, which is a typical mode of scanning for radars
used in operational networks. Calculation of the QVP requires horizontal homogeneity of the observed atmospheric processes.
The height-vs-time format of QVP represents the general structure of the storm or its evolution. Note that this is a novel
application of the QVP data product and the interpretation of QVP polarimetric variables differs from that of PPI or RHI scans
due to the averaging used to construct them.
The QVP input is used in this work to build a hierarchical structure based on the identified clusters and deliver an optimal
number of clusters based solely on internal properties of the multivariate polarimetric data. The optimal clustering is then used
to label the hydrometeor classes and to analyse the temporal evolution of the labelled microphysical processes.
The paper is organised as follows: in Section 2 we introduce the clustering methods we employ, Section 3 contains a description
of the polarimetric radar data and their processing, Section 4 describes the iterative clustering approach, leading to the

**Deleted:** and

**Deleted:** 2019

development of the hierarchical structure, Section 5 is devoted to the characterization of the clusters and their labelling, and Section 6 concludes with a summary, discussion, and thoughts on further perspectives.

## 2 Background of employed methods

The proposed hierarchical clustering algorithm identifies the optimal number of groups of data points (clusters) in a recursive loop and organizes the clusters in a hierarchical structure (undirected weighted graph). The two main steps of this approach are the cluster identification and the optimality check. The cluster identification is achieved after performing dimensionality reduction by principal component analysis followed by spectral clustering. The optimality check uses validity indexes to identify the final set of clusters, which best classifies the provided set of data. The description of the dimensionality reduction and clustering methods with background information about the validity indexes employed can be found in this section directly after a short introduction to the hierarchical clustering.

### 2.1 Hierarchical clustering

Hierarchical clustering is a type of clustering technique that splits or combines the data through an iterative process. Unlike "flat" clustering techniques, hierarchical clustering is not performed in one stage. Rather, it repeats the clustering process iteratively and keeps the information about each iteration of clustering in a hierarchical structure. In general, for a given set of multivariate data points, a hierarchical clustering algorithm, depending on top-down or bottom-up direction, either partitions (divides) or merges (agglomerates) the data into groups (a set of clusters) where data points assigned to the same cluster show similarity in multivariate values (depending on the context, it could be, for example, having a small distance to each other if the points are in Euclidean space). The direction of the process (top-down or bottom-up) may be chosen and depends on the number of individual points in the multivariate dataset and the needs of the underlying problem.

The top-down method begins with all available data points organized in one cluster and splits this cluster into subclusters until a certain criterion is reached or only solely singleton clusters of individual points remain in the set. The bottom-up method, on the other hand, begins with all points assigned to individual clusters and at each step, merges the most similar pairs of clusters into one until all the subclusters are agglomerated into a single cluster.

The optimal number of clusters in both approaches can be identified using a termination criterion. The hierarchical structure in the bottom-up approach needs to be completely finished before the optimal number of clusters can be identified otherwise the upper part of the tree will remain unknown. The top-down approach allows for the iterative process to be stopped at any point whilst preserving the upper part of the hierarchical structure. Another advantage of the top-down approach is the possibility to have more than two subclusters belonging to one parenting cluster. This allows an optimal number of subclusters for each parenting cluster, representing the data-driven inheritances in the resulting hierarchical structure. Although this advantage is often not used, and the bottom-up methods are preferred (Grazioli, 2015 and Rimbaud et al., 2019), the method presented here fully exploits it for the identification of an optimal number of subclusters in each iteration.

**2.2 Eigenvectors and Principal Components**

Principal component analysis (PCA) is a statistical technique mostly utilized in exploratory analysis of multivariate data. It extracts the most important information from the multivariate dataset generating a simplified view of the original data by dimensionality reduction.

To reduce a dataset's dimensionality a set of new orthogonal, non-correlated variables called principal components is calculated as linear combinations of the original variables. The first component is selected having the largest possible variance, so it best represents the diversity of the given data. The second component is generated under the assumption of orthogonality to the first component whilst also having the largest possible variance. This process is continued until the number of principal components is equal to the number of original variables ($d$). These components are exactly the eigenvectors of the correlation matrix and are employed as a basis for a new coordinate system (Wold, 1976; Abdi and Williams, 2010). The first $q$ calculated coordinates having satisfactory representativeness (e.g. 85 %) can be used to preserve the most important characteristics of the original data. These $q$ principal components can replace the initial $d$ variables ($q < d$), and the original data set, consisting of $N$ measurements on $d$ variables, is reduced to a data set consisting of $N$ measurements on $q$ principal components.

**2.3 Clustering method**

One of the most popular clustering methods is the k-means algorithm (Steinhouse, 1956). Through its simple interpretability, it is often used either as a single method or as a part of more computationally expensive clustering methods (e.g. Gaussian mixtures or spectral clustering). As a single method, it has difficulties with non-convex clusters and is known to perform poorly if the input variables are correlated (von Luxburg, 2007). As a basis of a more complex method (e.g. spectral clustering) it allows a solution of non-linear cluster shapes to be found (any low-dimensional manifolds of high-dimensional spaces).

The input data herein are multi-dimensional and were found to have non-convex cluster shapes, therefore the spectral clustering method was applied (Shi & Malik, 2000; Ng et al., 2002; von Luxburg, 2007). It works by approximating the problem of partitioning the nodes in a weighted graph as an eigenvalue problem of eigenvectors described above and by applying the k-means algorithm to this representation in order to obtain the clusters. This work implements the Ng et al. (2002) approach and analyses the eigenvectors of the normalized graph Laplacian.

Spectral clustering has several appealing advantages. First, embedding the data in the eigenvector space of a weighted graph optimizes a natural cost function by minimizing the pairwise distances between similar data points and such an optimal embedding is analytically deducible. Secondly, as it was shown in von Luxburg (2007), the spectral clustering variants arise as relaxations of graph balanced-cut problems. Finally, spectral clustering was shown to be more accurate than other clustering algorithms such as k-means (von Luxburg, 2007).

**2.4 Determining optimal number of clusters**

Clustering algorithms can be roughly divided into two groups based on whether the number of clusters to be found is predetermined or undetermined. Spectral clustering is a rather flexible technique in the sense that it can be used with a relaxation (i.e. when the number of clusters to be found is provided) or without it (when the number of clusters is determined by the multiplicity of the eigenvalue 0). As the approach chosen here is not interested in a flat partitioning of the data, rather we want to determine hierarchical structures, the determination of the optimal number of clusters is important. To identify this optimal number of clusters two evaluation scores are used in our method: the Wemmert-Gançarski (WG) index (Hämäläinen et al., 2017) and Bayesian Information Criterion (BIC) index (Pelleg and Moore, 2000; Hancock, 2017). The WG index was chosen as best performing according to comparison studies (Niemelä et al., 2018 and Hämäläinen et al., 2017). The BIC is best for the calculation of the posterior probability of a clustering. While the exact use of the indexes is described in Sect. 4, the WG and BIC indexes can be defined as follows:

Let the data set $X = \{x_i \in \mathbb{R}^d : i = 1, \ldots, N\}$ have clustering $C_K = \{c_k : k = 1, \ldots, K\}, (K < N)$, where $n_k$ – number of samples/points in the cluster $c_k$ and $I_{c_k}$ – indexes of the points in $X$ belonging to the cluster $c_k$.

1) For the WG index: Let $R_{(x_i)}$ represent the mean of relative distances between the points belonging to the cluster $c_k$ and the centre of its barycentric weight $g_k$. The $R_{(x_i)}$ value is calculated for each point $x_i$

$$R_{(x_i)} = \frac{\|x_i - g_k\|}{min_{k \neq k\prime}\|x_i - g_{k\prime}\|},$$

after that the WG–index

$$WG_X = \frac{1}{N}\sum_{k=1}^{K} max\left\{0, n_k - \sum_{i \in I_{c_k}} R_{(x_i)}\right\}, \quad (1)$$

is calculated representing the WG-index for the set $X$ of points partitioned into $K$ clusters (Desgraupes, 2017). The WG-index is a measure of compactness based on the distances between the points and the barycenters of all clusters.

2) For the BIC index:

Let us model each cluster $c_k$ as a multivariate Gaussian distribution $N_{(\mu_k, \Sigma_k)}$, where $\mu_k$ can be estimated as the sample mean vector and $\Sigma_k = \frac{1}{d(N-K)}\sum_{k=1}^{K}\sum_{i \in I_{c_k}}\|x_i - g_k\|$, can be estimated as the sample covariance matrix. Hancock (2017) showed that the optimal clustering is presented by maximum

$$BIC_{(C_K)} = \sum_{k=1}^{K}(n_k \log\frac{n_k}{N} - \frac{d\,n_k}{2}\log 2\pi\Sigma_k - \frac{n_k-1}{2}d) - \frac{1}{2}K_{(d+1)}\log N. \quad (2)$$

The BIC is an estimate of a function of the posterior probability of a clustering being true, under a certain Bayesian setup, so that a higher BIC in (2) means that a clustering is considered to be more likely to be the optimal clustering.

**3 Data and processing**

This section presents a description of the polarimetric radar data used by the hierarchical algorithm in this study, and some details of the data pre-processing that is applied before the QVP calculation-processing. Note that the method presented here is generally applicable with similar multivariate data from other sources. In addition, in situ observations from the FAAM Airborne Laboratory (FAAM BAe-146) are presented in this section. These data will be used for assigning the labels of the hydrometeor classes to the detected clusters or cluster groups in the radar observations.

**3.1 X-band radar observations**

The polarimetric data employed to demonstrate the method developed in this study were collected by the NXPol radar whilst it was located at the Chilbolton Atmospheric Observatory (CAO), part of the UK's National Centre for Atmospheric Science's Atmospheric Measurement and Observation Facility (AMOF), in southern England (Lat. 51.145° N, Long. 1.438° W ) from November 2016 to May 2018 (Fig. 1). The NXPol is a mobile Meteor 50DX (Leonardo Germany GmbH) X-band, dual-polarization, Doppler weather radar with a 2.4 m diameter antenna. The radar is a magnetron-based system and operates at a nominal frequency of 9.375 GHz ($\lambda \sim 3.2$ cm). The detailed characteristics of the NXPol radar can be found in Neely et al. (2018). From the observations made in 2017–2018 we selected eight dates with the longest precipitation events occurring within 30 km range of the radar presented on Fig. 1. The exact dates and the total number of volume scans per date can be found in Table 1. Radar data selected for this study can be found in the list of mobile X-band radar observations on the CEDA archive (Bennett, 2020).

**3.2 Polarimetric variables and temperature data**

Here we chose to use the polarimetric variables $Z_H$ [dBZ], $Z_{DR}$ [dB], $\rho_{HV}$ [–], and $K_{DP}$ [° km$^{-1}$] as well as temperature $T$ [° C] to demonstrate the described clustering technique. The four polarimetric variables were selected as a subset of all the possible variables as they provide complementary information about the observed hydrometeor properties. Here $K_{DP}$ is calculated as the linear gradient of differential phase shift, where the phase shift has been filtered to remove non-meteorological targets ($\rho_{HV} > 0.85$) and progressively smoothed using decreasing length averaging windows and $Z_H$ and $Z_{DR}$ are corrected for attenuation using a linear correction (Bringi et al. 1990). Temperature was added to the set of input variables following the reasoning of similar studies in which either height relative to 0° C–isotherm (Grazioli et al., 2015) or the index representing ice- or liquid-phase of observed precipitation (Besic et al., 2016) was included. The full input vector used in this study can be represented as:

$$x = [Z_H, Z_{DR}, \rho_{HV}, K_{DP}, T], \quad (3).$$

Note that this does not preclude the use of differing sets of variables in future studies. The input data used here were pre-
processed before being utilized in the clustering algorithm: all range bins that were located at distances less than 400 m from
the radar were removed from all input variables data to reduce the influence of side lobe noise.
The temperature data were taken from the Met Office Unified Model (UM) and interpolated onto the polar grid of the radar's
observations. The original data can be found on the CEDA archive (Met Office, 2016). Past assessments of the accuracy of
these temperature values suggest that the gridded temperatures are within 1° C of co-incident profiles measured by radiosondes
except in the case of strong inversions or frontal boundaries. An example of one day's observations (2017-05-17) represented
in the height-vs-time format of the QVP of four polarimetric variables with the temperature presented as isotherms is found in
Fig. 2. Similarly, other dates from our list of cases (Table 1) are in Fig. B1.
**3.3 QVP and thresholding**
QVP of the input variables are obtained as the azimuthal average of the data from a standard plan position indicator (PPI) scan
at 20° antenna elevation angle (Ryzhkov et al., 2016). The 20° PPI is the highest of ten PPIs of the volume scanning strategy
used by NXPol which starts the scanning from 0.5° elevation angle. The use of 20° PPI minimizes the effects of radar beam
broadening and horizontal inhomogeneity. The beam broadening effect becomes dominant at higher altitudes when observed
by low elevation scan as was shown in Ryzhkov et al. (2016). The radar beam of 1° opening at 20° elevation is about 100 m
at 2-km height, 240 m at 5-km height, and reaches almost 480 m at 10-km height. The resulting profiles have 197 voxels in
each QVP at the altitudes between 0 and 10 km above mean sea level (AMSL) and covering about 30 km range from the radar.
It was also shown in Ryzhkov et al. (2016) that the decrease of $Z_{DR}$ from the oblate spheroidal hydrometeors at 20° elevation
is within the common measurement error of $Z_{DR}$ (0.1–0.2 dB).
An advantage of QVP is that they reduce statistical errors within the input dataset while the height-vs-time format of QVP
naturally represents the temporal dynamics of microphysical processes observed in the radar data. To ensure the observations
are representative of large-scale meteorological features that may be averaged together, the QVP voxels are used in the analysis
only if more than 270 of the 360 azimuthal bins at the range in the PPI scan contain valid data.
**3.4 In situ Observations**
For the labelling of the clusters, in situ observations can be used to assess any of the clusters within the hierarchical structure
produced by the clustering algorithm. This allows for flexibility in the granularity used to examine the observations. In this
study, the in situ FAAM BAe-146 observations are used to demonstrate the labelling and assessment of the final set of clusters.
The FAAM BAe-146 is a publicly funded research facility that, as part of the National Centre for Atmospheric Science
(NCAS), supports atmospheric research in the United Kingdom by providing a large instrumented atmospheric research aircraft
(ARA) and the associated services. The ARA is a modified British Aerospace 146-301 aircraft. Further details of the FAAM
BAe-146 aircraft instrument systems are available at https://www.faam.ac.uk/the-aircraft/instrumentation/. In situ data for this
study comes from FAAM BAe-146 flights C013, C076, C081, and C082 (FAAM, 2017; FAAM, 2018a; FAAM, 2018b;
FAAM, 2018c) and observational data are available on the CEDA archive. The dates of the flights with corresponding flight
numbers can be found in Table 1.
For the cases examined, FAAM BAe-146 was equipped with two Cloud Imaging Probes (CIP), which are manufactured by
Droplet Measurement Technologies and described in Baumgardner et al., 2001. The CIP's are mounted underneath the aircraft
wings and provide 2-bit grayscale images of cloud particles as they pass through the instrument sample volumes. Each CIP
houses a 64 elements photodiode detector, with one CIP having an effective pixel size of 15μm (referred to as CIP15), and the
other having an effective pixel size of 100μm (referred to as CIP100). Therefore, the CIPs provide images of particles in size
ranges of 7.5 μm to 952.5 μm for CIP15 and 50 μm to 6350 μm for CIP100. All probes have 'Korolev' anti-shatter tips – the
width of which is 70 mm for the CIP100 and 40 mm for the CIP15.
Particle size distributions are calculated based on CIP data where particle size is defined as being the maximum recorded length
in either the axis of the detector array (X) or along the direction of motion (Y). All particles with inter-arrival times $< 10^{-6}$ s
are rejected as indicative of shattering as in Field et. al. (2006). The centre-in approach (Heymsfield and Parrish, 1978) for
estimation of particle concentrations from the sample volume is used to calculate the size of partially imaged particles. It
should be noted that despite using the centre-in method with the CIP data, which increases the effective sample volume for
larger particles at the expense of uncertainty in particle size, the ability to measure particles with size > 6 mm is negligible
with this configuration. An indication of the potential presence of such large particles can be obtained through a visual
inspection of the particle images, but no conclusions can be drawn. Also, there are significant uncertainties associated with the
derived properties from the CIPs, which are of the order 20% for number-based properties (Baumgardner et al., 2017). In our
analysis we are not concerned with absolute concentrations from the CIPs. Instead, we are using the CIP data in a qualitative
manner to provide a general framework for comparison with the HC results obtained from the radar observations.
As we base our clustering on the QVP of radar observations, which at each range from the radar are averaged over all available
azimuths, a direct comparison to aircraft observations taken at an exact position and timestamp would not make a representative
comparison. Thus, for the comparison, we have selected the 20-second intervals from the CIP15 and CIP100 data that
correspond to the spatial domain and times of the individual 20° PPI scans that are used to create the QVP. Over these 20 s
intervals the mean number concentrations per particle size bin are calculated (Fig.10). Fig. 9 presents examples of particle
imagery from the CIPs, which typically represents less than 1s of the total 20s of data and shows derived properties of the
particles over the entire 20s sampling period when the airplane observed the atmosphere over the QVP domain.
In order to provide insight into the nature of the particle imagery, we have separated the particle concentrations in Figure 10
into three categories, two of which are based on an analysis of the particle shapes, and another category for partially imaged
particles. "Round" particles are those which have a circularity between 0.9 and 1.2, and particles with a larger circularity are
labelled as "Irregular" - this gives a rough separation into particles which are likely to be liquid water vs ice (Crosier et al.,
2011). "Edge" particles are those which are only partially imaged, as indicated by pixels at the extreme edge of the array being
triggered. For the particles which are considered round, the particle size and subsequent concentration is corrected for out-of-
focus effects (Korolev, 2007). This out-of-focus correction is not applied to the "irregular" or "edge" images, as there is no

Deleted: . Particle

Deleted: are calculated

Deleted: using the centre-in approach (Heymsfield and Parrish, 1978)

1. evidence to show this is an appropriate correction to make. No attempts have been made to classify particle images, for two
2. key reasons. (1) The larger particles, which have the greatest influence on the polarimetric properties, are poorly sampled by
3. the CIPs. (2) A recent study by O'Shea et al. (2020, in review at AMTD) suggests existing procedures to classify particle
4. images using the CIP can lead to inaccurate results due to the effects of diffraction when particles are imaged more than a few
5. mm off the focal plane, which is a the most common scenario. A thorough assessment of the accuracy of these image
6. classification algorithms, with respect to particle size and probe configuration, is much needed.

7. **4 Clustering of QVP**

8. Here the clustering steps are described in general. A corresponding overview of the approach is provided in Fig. 3. The
9. proposed approach uses QVP voxels and temperature data that has been interpolated to the same volume. This data forms the
10. points of a 5-dimensional space ($d = 5$). The PCA (Sect. 2.2) reduces the number of dimensions to $q$. The $q$-dimensional data
11. are partitioned into $K$ clusters, where $K$ iteratively increases ($K = 2, 3, ...$) until the optimal number of clusters is reached,
12. according to the WG index Eq. (1) in Sect. 2.4. With each of the clusters achieved in this level ("Outer Loop" in Fig. 3), the
13. process is recursively repeated starting with the PCA calculation and continuing until the optimal partitioning of the sub-
14. clusters is reached ("Inner Loop" in Fig. 3). The total partitioning is confirmed with the BIC index Eq. (2) in Sect. 2.4. When
15. the BIC's local maximum is reached the partitioning is considered to be optimal. A detailed description of each of these steps
16. can be found in the following subsections and the code can be made available on request.

17. **4.1 Start of hierarchical clustering**

18. The hierarchical clustering starts with data standardization and dimensionality reduction of the original 5-variable input data
19. $X$ Eq. (3) into a $q$-dimensional dataset of principal components (Sect. 2.2). The non-parametric transformation based on the
20. quantile function maps the data to a uniform distribution. This standardization helps to deal with outliers and satisfy PCA data
21. assumptions.
22. To start the loop, all $N$ pixels of the input data are used. In later loops, only subsets of the original 5-variable data ($I_{c_k}$)
23. belonging to active cluster $c_k$ are processed in the Inner Loop (Fig. 3). The first $q$ principal components with the largest
24. variance, having in total at least 85 % representativity of the original dataset, are selected in this step.
25. The representativity threshold of 85 % was chosen arbitrarily as it reduces the initial 5-variable input space up to 3-dimensions
26. ($q \leqq 3$) in most cases, which effectively simplifies the clustering problem and does not influence the overall outcome. The
27. threshold can be reduced to further reduce the dimensionality, but it was found that this negatively influences the clustering
28. accuracy. A higher threshold will retain the high dimensionality of the original dataset but will slow down the clustering
29. process without gaining further information from the dataset.

**4.2 Iterative process to find the optimal number of clusters (Inner Loop)**

At the start of the hierarchical clustering, we begin directly with the first call of the Inner Loop (Fig. 3). The iterative process in the Inner Loop commences with all $N$ QVP pixels represented by the first $q$ principal components. The spectral clustering processes these input data starting with the number of clusters $K = 2$. The number of clusters increases ($K = 2, 3, \ldots$) with each cycle of spectral clustering within the Inner Loop, and at the end of each iteration the WG index Eq. (1) is calculated for the achieved clustering $C_K$. At the moment the local maximum is achieved in the WG index values, the clustering in which it was reached ($C_K'$) is accepted as the main cluster set of the current level of the hierarchical tree and these clusters become the set of active clusters ($A$). Set $A$ will be used in the Outer Loop of the implemented hierarchical algorithm. The active clusters detected in the first level of the hierarchical structure by spectral clustering for the data on 2017-05-17 are shown in Fig. 4.

**4.3 Optimal number of clusters for the total dataset (Outer Loop)**

In the Outer Loop of the hierarchical algorithm, the BIC index Eq. (2) is calculated for the active clusters produced by the Inner Loop (Fig. 3). If the BIC index is calculated for the first time (i.e. the start of the algorithm run, $j = 1$) or the BIC index values do not show any local maximum, the algorithm continues by calling the Inner Loop for each individual cluster from the set of active clusters ($A_j$) formed by the calls of the Inner Loop described above.

For the first level of hierarchical clustering all $C_K'$ are immediately accepted as active $A_1 = C_K'$. In the Outer Loop, after calculating the BIC, the original 5-dimensional data belonging to each cluster $c_k' \in C_K'$ are sent to the Inner Loop and clustering ($C_K''$) achieved by the Inner Loop is used to replace the cluster $c_k'$ in the set of active clusters $A_1$. If the BIC index calculated on this 'suggested set' shows that the clusters introduced to the $A_1$ increase the value of BIC, the suggested replacement is accepted and the set of active clusters is updated as $A_2 = C_K'' \cap C_K'/c_k'$. The Outer Loop then continues with the next cluster from the original set $A_1$. When the BIC value does not increase with the 'suggested set' the set of active clusters does not change, and the algorithm continues with the next cluster from the set $A_1$.

**4.4 Next recursion or finalization of results**

The final set of active clusters is reached when the value of the BIC index does not increase with any further suggested split in the current active set of clusters. At this level of detailization, the optimal clustering for the provided input data has been reached. For the QVP dataset described in Sect. 3, a final set of 13 active clusters is reached (see Fig. 5 and Fig. 6). The relations between these final clusters (f_cl1, ..., f_cl13) and the 3 parent clusters from the first Inner Loop run (Fig. 4, panel (a)) are shown in Fig. 5.

**5 Labelling**

Once the optimal number of clusters is determined and the hierarchical clustering structure is built, the clusters can be characterized by their centroids and labelled with appropriate hydrometeor classes using the available verification data. The clusters for which direct verification data are not available may still be labelled with an appropriate hydrometeor class based on the scattering characteristics described by the original polarimetric radar variables and considering their position in the hierarchical tree and height-vs-time QVP representation. As QVP polarimetric characteristics differ from polarimetric characteristics of hydrometeors observed by PPI and RHI scans, care must be taken when comparing these results to the literature. Labelling the obtained clusters can be performed for the different levels of granularity depending on the user's needs and interests. Note the purpose of the labelling shown here is to demonstrate the ability and validity of the technique rather than performing a rigorous study of the underlying microphysics observed. The latter will be reserved for follow-up studies utilizing this technique in a focused manner.

**5.1 Level by level clusters check**

From the visual verification of the first level parent clusters in Fig. 4 panel (a) and panel (b) we can deduce that there are two child clusters representing the upper/elevated (ice dominated root.cl2) and the lower (water dominated root.cl1) parts. The second level clusters from the second loop (panels (c) and (d)) show a well-identified "bright band", belonging to the melting layer (ML), (root.cl2.cl2) and a main solid-phase cluster (root.cl2.cl1), both belonging to the cluster representing the ice-phase dominated part (root.cl2) of the QVP (panel (b)). The three child clusters of the parent root.cl1 cluster (panels (c) and (d)) are the two rain-type clusters (root.cl1.cl2 and root.cl1.cl3) below the "bright band" and cluster root.cl1.cl1 with most points located above the "bright band". In further loops, the main ice-phase cluster (root.cl2.cl1) is split into nine child clusters (Fig. 5) and examples of their positioning in time-vs-height format of QVP can be observed in Fig. 6 or Fig. B1.

**5.2 Characteristics of the clusters**

The 13 final clusters can be characterized by their centroids (Fig. 7) or their relevant statistics (Fig. 8 and Table A1). The centroid characteristics in Fig. 7 are plotted as spider plots where each of the five variables is represented by an azimuthal axis. The filled pentagons in each subplot represent the cluster's centroid in the five-variable space based on all the data available in this study. Each vertex of the pentagon shows the centroid's value in one of the five variables. The non-solid lines in the subplots of Fig. 7 represent the centroids of the same cluster but based solely on the data from one of the eight cases (Table 1).

Figure 7 confirms a distinction made at the first and the second cycles of the Outer Loop (Fig. 4) between three types of clusters: liquid-phase clusters (f_cl1, f_cl2 and f_cl3), having lower $K_{DP}$ and warmer $T$ values, ice-phase clusters (f_cl4, f_cl5, f_cl6, f_cl7, f_cl8, f_cl9, f_cl10, f_cl11), and f_cl12 all with more pronounced $\rho_{HV}$ values; and a very different looking f_cl13, having warmer $T$ and rather low $\rho_{HV}$ values.

The largest differences between the centroids selected from the total dataset and the centroids corresponding to the eight
considered cases occur in the temperature (T) values, especially for the clusters f_cl2, f_cl3, f_cl7, f_8, f_cl10 and f_cl12.
These variations can be explained by the origin of the temperature data, which are estimates from the NWP model and do not
always correctly represent the real situation.
The next variable with a rather large variation in several clusters is $K_{DP}$. In part this variation may be due to the fact that
this variable has an extremely skewed distribution. Clusters f_cl1, f_cl7, f_cl11, f_cl12 and f_cl13 have the highest
variation in $K_{DP}$ values between the centroids calculated on different cases (Fig. 7). As $K_{DP}$ can be influenced by the amount
of ice/water particles in the atmosphere, it might be that the clusters have variations in the number of particles. This hypothesis
can only be verified with FAAM BAe-146 observations of the same cluster on different dates. Unfortunately, such verification
is not possible for all clusters and more in situ observations are required.
From the variations of centroid values in the five input variables in Fig. 7, we can also see that the main liquid-phase cluster
(f_cl2) has rather different characteristics in different cases. Case to case it shows large variations in $Z_H$, $K_{DP}$, $\rho_{HV}$, $T$ and the
highest mean temperature value (6.8° C) among all other clusters. As observed in the histogram of percentages of the cluster
points in Fig. 8 panel (f), f_cl2 is rather big (13.7 % of the total number of points), but does not have the highest percentages
of points in all eight analyzed cases, only in the 2017-05-17 case (9.3 % compared to ≤ 1.5 % in other cases) (Fig. 7, histogram
per case). Combining all these aspects together we deduce that f_cl2 includes rain of varying intensities and different drop
sizes.
The other rain cluster f_cl3 has less variability in centroid values and has ~20 dBZ smaller $Z_H$ values than f_cl2 probably due
to the smaller drop sizes in this cluster. f_cl3 has the smallest mean $Z_H$ (7.18 dBZ) and mean $K_{DP}$ values (-0.097 ° km$^{-1}$) of
all "water" clusters (f_cl1, f_cl2 and f_cl3). This cluster is often observed at the beginning and at the end of the storm in height-
vs-time format representation of the optimal clusters (Fig. B1) and is labelled as "light rain".
Almost all centroids in Fig. 7 have no or very limited variation in $\rho_{HV}$ or $Z_{DR}$ values except for f_cl13. This cluster has no
variation in temperature (T), having all centroids at 0° C. As such, f_cl13 corresponds to the area in the data referred to as a
"bright band". According to the box and whiskers plots in Fig. 8 cluster f_cl13 has the highest mean $Z_H$ (24 dBZ), mean $Z_{DR}$
(0.99 dB), and the lowest mean $\rho_{HV}$ (0.93) compared to the other optimal clusters and it is mostly located near 0° C. These
characteristics immediately indicate that f_cl13 can be labelled as the "bright band" belonging cluster having mixed-phase
(MP) particles.
The MP cluster (f_cl13 in Fig. 6) is observed to have some sagging areas: between 10:00 (UTC) and 12:20 (UTC), around
16:00 (UTC) and near 18:00 (UTC). Note that f_cl1 is observed above the MP cluster f_cl13 exactly at these time intervals
(Fig. 6). This sagging "bright band" signature is often observed where aggregation and riming processes are occurring directly
above the melting layer (Kumjian et al., 2016 and Ryzhkov & Zrnic, 2019). This suggests that f_cl1 can be associated with the
processes of aggregation or riming and labelled accordingly.

Deleted: " of the ML.

Deleted: 8 the "bright band"

Deleted: "bright band"

Deleted: "bright band"

Deleted: ML

Deleted: and

Looking at the percentage of points belonging to each cluster in the optimal clustering set (Fig. 8, panel (f)), we see that clusters f_cl7 and f_cl12 have less than 2 % of points and most probably represent some sporadic and/or special conditions. These clusters also have been separated early from the other "low ice" (f_cl4, f_cl5, f_cl6) and "elevated ice" (f_cl8, f_cl9, f_cl10, f_cl11) clusters and are located near the top of the hierarchical tree (Fig. 5). Both clusters f_cl7 and f_cl12 have smaller absolute mean $Z_{DR}$ values (-0.062 dB and 0.097 dB correspondingly) than the other "ice" clusters (Fig. 8, panel (b).). f_cl7 is also characterised by the highest mean $K_{DP}$ value (0.44 ° km$^{-1}$) among all of the clusters (Fig. 8, panel (d)). The combination of rather high $Z_H$ (17 dBZ) and high $K_{DP}$ at temperatures around -15 °C indicates a cluster with high particle number concentration of small ice crystals mixed with a small amount of bigger aggregates. This cluster is potentially a manifestation of the rapid growth of ice via vapor deposition and onset of aggregation in the dendritic growth layer (DGL) discussed in the details in Griffin et al. (2018). These characteristics were also recognised as a signature of dendritic crystals in Bechini et al. (2013). f_cl12 has the lowest mean $Z_H$ value (-3 dBZ) from all optimal clusters. Combining the low mean $Z_H$ with low mean $Z_{DR}$ (0.097 dB) and temperature about 3 °C we can assume that f_cl12 can be labelled as small droplets (i.e. drizzle).

f_cl11 belongs to the "elevated ice" clusters and in most cases (see Appendix, Fig. B1) appears as a column in the height-vs-time representation (around 07:00 UTC in Fig. 6 or in Fig. B1 panels (a) 07:00–08:00 UTC; (c) 05:00 UTC and 07:30 UTC; (g) 12:00 UTC; (h) 12:00–12:30 UTC) filling all the altitudes from the top of the cloud to the ML. This cluster has low mean $Z_H$ value (3 dBZ), one of the highest mean $Z_{DR}$ values (0.92 dB) and a close to zero mean $K_{DP}$ value (-0.009 ° km$^{-1}$). f_cl11 can be labelled as the pristine ice crystals class, as they typically have high aspect ratios ($\gg$1) and tend to fall preferentially with their major axis aligned horizontally (Keat and Westbrook, 2017).

Clusters f_cl8, f_cl9 and f_cl10 belong to the "elevated ice" branch of the hierarchical tree (Fig. 5). Among these clusters, f_cl9 has the most different characteristics compared to f_cl8 and f_cl10 clusters. f_cl9 has higher mean $Z_H$ (11 dBZ) and $Z_{DR}$ (0.59 dB) in combination with a lower $\rho_{HV}$ (0.97).

f_cl8 and f_cl10 have rather similar characteristics to each other (Fig. 7 and Table A1). The small difference between these two clusters is in a higher mean $Z_H$ (8.6 dBZ) and $K_{DP}$ (0.14 ° km$^{-1}$) for f_cl8 compared to 5.8 dBZ and 0.028 ° km$^{-1}$ for f_cl10. Both clusters are the main "elevated ice" clusters. f_cl10 has a warmer mean temperature (-8.2° C compared to -16.7° C for f_cl8) and most of the time is located near f_cl8 at the beginning or at the end of the observed event (Fig. B1).

The main "low ice" clusters are f_cl4, f_cl5 and f_cl6. Clusters f_cl5 and f_cl6 are often observed together with f_cl6 located above f_cl5. The f_cl4 has several appearances in height-vs-time formats of events (see Appendix, Fig. B1, e.g. 09:00 UTC panel (a); 11:00 UTC panel (b); 09:00 and 17:00-18:00 UTC panel (e)), mostly above f_cl1 reaching higher altitudes in the data.

Measurement errors may influence the clustering results. As it was shown by Bringi et al. (1990) noise in the observations has a strong impact on k-mean HCA results. Unfortunately, it is impossible to run the same type of analysis conducted by Bringi et al. (1990) for an unsupervised hierarchical clustering algorithm as the added noise might deliver modified hierarchical structure with another optimal number of clusters and direct comparison to the original set of final clusters would be

Deleted: potentially

Deleted: low particle number concentration of very

Deleted: ice crystals.

Deleted: Areas with similar characteristics in the height-vs-time representation of the QVPs were labelled as dendritic growth layer (DGL) by Ryzhkov et al. (2016) and Trömel et al. (2019).

Deleted: Of course, not all clusters can be labelled with absolute confidence and in situ observations can help to verify these initial suppositions.¶

impossible. Here this issue of noise is partially addressed through our use of QVP. In particular, azimuthal averaging of a QVP
reduces the noisiness of the differential phase within the melting layer (Trömel et al. (2013, 2014)) and was recommended in
Kumjian et al. (2013) to quantify rather small enhancements of $Z_{DR}$ and $K_{DP}$.
The mean negative values of $Z_{DR}$ and $K_{DP}$ in some clusters (f_cl1, f_cl3 and f_cl7) might point at potential biases due to
miscalibration of $Z_{DR}$, differential attenuation, or backscatter differential phase in the melting layer. Biases in the data
such as miscalibrations will not impact the clustering process but will impact the labelling as it is based on cluster
characteristics. A miscalibration of $Z_{DR}$ can also be excluded as we routinely perform calibration of this variable. $Z_H$ and
$Z_{DR}$ are corrected for the attenuation in the data pre-processing. Biases caused by backscatter differential phase in the
melting layer (Trömel et al., 2014) have not been removed and are evident in the cluster characteristics. The influence of the
backscatter differential phase needs further investigation. As discussed, not all clusters can be labelled with absolute confidence
solely based on the cluster's characteristics and in situ observations can help to verify these initial suppositions.
5.3 Clusters versus in situ observations
**5.3.1 In situ data**
For the verification of the preliminary labelling made in Sect. 5.2, data from the CIP15 and CIP100 on board the FAAM BAe-
146 is utilised. FAAM BAe-146 aircraft flights were performed on four out of the eight days (Table 1) of radar observations:
2017-05-17, 2018-01-24, 2018-01-24 and 2018-02-14. The flight altitudes and the timestamps when the aircraft was inside the
QVP domain can be observed on the height-vs-time representations of the optimal clusters in Fig. B1.
Out of the four available flights, there are 23 periods of 20-second intervals which result in a total of 460 seconds of flight
time when the aircraft was inside the QVP domain and a cluster can be assigned to the corresponding height. Of these 23
periods there are observations corresponding to 9 unique clusters (f_cl1, f_cl2, f_cl3, f_cl4, f_cl5, f_cl6, f_cl8, f_cl10, f_cl12).
This samples 70 % of the final clusters. From these time series, we present examples of CIP15 and CIP100 images for each
cluster (Fig. 9) and mean particle size distributions of the data observed during the 20-second interval (Fig. 10).
**5.3.2 Liquid-phase clusters**
The liquid-phase clusters f_cl1, f_cl2 and f_cl3 correspond to in situ data which contains relatively high concentrations (>1
000 m$^{-3}$) of small particles (mostly < 200 μm in size) which appear round (Fig. 9 and Fig.10, panels (a)–(c)). This strongly
supports the idea of large amounts of liquid water being present in the cloud, which supports the labelling of f_cl1-f_cl3 as
being influenced by liquid water hydrometeors. When looking in detail at CIP100 in situ observations, and to some extent
CIP15 observations at sizes > 200 μm, we can see some significant difference between f_cl1-f_cl3, which we will now discuss.
f_cl1, observed above the "bright band", was previously assigned to be the result of aggregation and riming. In this region, the
CIP100 (Fig. 9, lower part of panel (a)) shows particle imagery and particle size distributions segregated by shape which show
the presence of large ice particles (Fig. 10, panel (a)), again confirming the previous cluster labelling. The CIP100 data shows
the presence of irregularly shaped particles, ranging in size from ~1-4 mm, with concentrations in each bin of the order 1 m$^{-3}$.
This suggests a mode of snow particles is present at the same time as the previously mentioned liquid droplet mode. Many
small water droplets in the CIP15 observations (Fig. 9 and Fig.10, CIP15 of panel (a)) could indicate either the presence of
warm cloud processes or small ice crystals melting first around the ML. The second interpretation is supported by the imagery
from the CIP100 which suggests melting has not started to occur on the larger particles. In this case, the larger aggregate
snowflakes fall to lower altitudes before they start to melt and form the clear "bright band" in the QVP.
f_cl2 was characterized by the strong variation in $Z_H$, $K_{DP}$, $\rho_{HV}$, and $T$ of the cluster's centroids in different cases. Both CIP15
and CIP100 have small round shape particles in the corresponding images (Fig. 9, panel (b)). The mean concentrations per
particle size distributions (Fig. 10, panel (b)) show the prevalence of particles recognised by shape as water droplets. The
droplets of < 2 mm size have the occurrences of the order from 10 m$^{-3}$ to 90 000 m$^{-3}$ with higher orders corresponding to
particle sizes < 200 µm. Summing up previous analysis and in situ observations we can assign f_cl2 to a "liquid" cluster, which
includes rain of varying intensities and different drop sizes.
f_cl3 also has predominantly small round shape particles (mean size $\mu$ = 128 µm) in the CIP15 panels (Fig. 9 and Fig.10,
upper part of panel (c)). The CIP100 data (Fig. 9, lower part of panel (c)) were not processed due to technical issues with the
probe so water/ice concentrations based on this data are unfortunately not available. The high concentrations (1000-5000 m$^{-3}$)
of small size (< 200 µm) particles are assigned to water (Fig.10, panel (c)). Concentrations of the larger particles (> 200 µm
and < 800 µm) are very low (< 100 m$^{-3}$). Considering these observations, the cluster's characteristics and the fact that the
cluster appears mostly in the beginning or at the end of the events (Fig. B1) we can assume that the cluster either represents a
very light rain/drizzle or indicates a partially filled QVP domain in the original data.

### 5.3.3 Solid-phase clusters

The CIP15 images for f_cl4 show a mix of larger irregularly shaped particles (aggregates of snowflakes) and relatively few
tiny ice crystals (Fig. 9, panel (d)) having very small irregular shapes. The mix of small (< 1 mm) particles recognised as water
and ice has low total concentrations (500-800 m$^{-3}$; Fig.10, upper part of panel (d)). Particles of larger sizes (> 1 mm and < 3.0
mm) have concentrations from 100 m$^{-3}$ to 800 m$^{-3}$ (Fig.10, lower part of panel (d)) and were recognised as ice due to their
irregular shapes. The small number of "round" particles recognized as liquid are likely an artefact of the data processing due
to out-of-focus imaging of the numerous ice particles which are present - such artefacts appear when particles are observed at
the edges of the depth of field (O'Shea et al, 2019). Accordingly, this cluster can be assigned to the mix of pristine ice and
some formed aggregates, all having low concentrations.
f_cl5 and f_cl6 show very similar images in CIP15, but the CIP100 images illustrate the difference between these two clusters
(Fig. 9 panels (e) and (f)). The shape analysis of CIP100, suggests both clusters include a low concentration (40-50 m$^{-3}$) of
small round shape particles (< 1 mm size; the lower part of panels (e) and (f), Fig. 10). Similar to f_cl4, the number of liquid
particles are likely an artefact of the data processing (O'Shea et al, 2019). The main difference between these clusters is shown
in the part of the data recognized as ice. The mean size of the particles of f_cl5 is about 1.7 mm and has occurrences of order
100 m$^{-3}$ (Fig. 10, panel (e)). While the mean size of particles in f_cl6 is a bit smaller - 1.4 mm and particles around that size
have a higher occurrence of order 400 m$^{-3}$ (Fig. 10, panel (f)). This difference between f_cl5 and f_cl6 resembles the
aggregation processes when dendritic crystals of higher concentrations formed at higher altitudes (f_cl6) start to clump together
during their fall and form aggregates (f_cl5) with a lower concentration of particles.
Clusters f_cl8 and f_cl10, belonging to the "elevated ice" branch of the hierarchical tree (Fig. 5), are also represented by the
very similar images of CIP15 and CIP100 observations (Fig. 9, panels (g)–(h)) with the difference in the particles'
concentration (Fig. 10, panels (g)–(h)). f_cl8 has a higher concentration of small size (<1 mm) particles (up to 1000 m$^{-3}$)
recognised as spherical than f_cl10 (up to 300 m$^{-3}$). The bigger particles captured in CIP100 corresponding to f_cl8 are of
bigger size (from 1 mm and up to 3 mm) and also have a higher concentration (< 500 m$^{-3}$) compared to particles of f_cl10
having a maximum size of about 2 mm with concentrations < 200 m$^{-3}$. The example of CIP100 images suggests that these
particles are dendritic in nature (Fig. 9, panel (h)). Again, similar to the artefacts discussed when looking at the CIP
observations for f_cl4, f_cl5 and f_cl6, there is likely an erroneous classification of small size particles (O'Shea et al, 2019).
Thus, according to in situ data, f_cl8 can be assigned to a mix of pristine ice and bigger aggregates and f_cl10 to a low
concentration mix of pristine ice and smaller aggregates.
The last in the list of clusters verifiable with the FAAM BAe-146 data is f_cl12 (Fig. 9, panel (i)). The lower panel (i) of Figure
10 shows occurrences of order 10 000 m$^{-3}$ for very small particles up to 150 μm and occurrences of order 1000 m$^{-3}$ for the
droplets of sizes between 150 μm and 250 μm with almost no occurrences of bigger particles. Unfortunately, concentrations
from CIP100 data are not available for this cluster due to technical issues in the CIP100 probe. f_cl12 contains the fewest
number of data points in QVP analysis (Fig 8. panel (f)) and there are no other in situ observations related to this cluster. Based
on available data, we could assume that the cluster has a high concentration of tiny water droplets (potentially drizzle). On the
other hand, similar to the data for other classes, the CIP analysis may have misclassified these as liquid due to their small round
appearance in the observations when in reality the observations could represent high concentrations of small ice particles.
Likewise, the mean temperature of this cluster is close to 0° C, so no definitive label may be given based on the observations.
Thus, the physical interpretation of this cluster is ambiguous though the cluster is separate within the multivariate space.
**5.3.4 Assignment conclusions**
The rest of the clusters need to be assigned by means of human interpretation according to the cluster characteristics or
deduction from the interactions and temporal evolution of already assigned clusters. The summary of the assigned clusters can
be found in Table A1 of the Appendix A. Application of in situ observations for the assessment of QVP-based clusters has its
limits as not all optimal clusters were captured by the FAAM BAe-146 flights and this process requires a comparison of data
from essentially one-point measurement to the cluster based on the mean QVP domain values. An appropriate validation

Deleted: as

Deleted: particles

Deleted: mix

process would utilise columnar vertical profiles (CVP) as described in Murphy et al. (2018) with the thorough collocation of
the aircraft observations. Utilising CVPs within the presented technique is a part of the planned work for the future.
**6 Summary and Conclusions**
This paper presents a new technique of hydrometeor classification from QVP. Note that both the data-driven approach and the
use of QVP is novel. In this technique, the hydrometeor types are identified from an optimal number of hierarchical clusters,
obtained through a recursive process. This recursive process includes an initial dimensionality reduction by principal
component analysis followed by spectral clustering. Spectral clustering performed in the PCA space allows us to identify
clusters that would have a non-convex form in the original multivariate input space. This property of the algorithm makes it
unique and advantageous in comparison to other classification methods, which separate classes by hyperplanes.
The final set of clusters is identified with an optimality check using validity indexes. This represents the first attempt, in top-
down hierarchical clustering of weather radar data, to identify the number of clusters based solely on the embedded data
characteristics. This data-driven technique produces an optimal number of clusters and keeps the hierarchical structure built in
the clustering process. The final set of clusters may be labelled based on their positioning in the hierarchical structure, the
characteristics of their centroids and co-incident in situ observations. Depending on the user's needs and interests, the labelling
can also be performed for different levels of granularity. In the example shown in this study, we utilise observations collected
during several FAAM BAe-146 flights to demonstrate the advantages this technique has in the labelling process. In this case,
based on the data available, 70 % of the clusters were labelled using the coincident CIP observations. The other 30 % of the
clusters, which were not sampled during the FAAM BAe-146 flights of this study, were labelled based on the cluster
characteristics, their positioning in the hierarchical structure and considering interactions with clusters in a height-vs-time
format of original QVP data.
Thus, in this study, we find that a data-driven HC approach is capable of providing an optimal number of classes from the
observations. Moreover, the embedded flexibility in the extent of granularity is the main advantage of the technique. Each
branch of the hierarchical structure can be cut out at any level and the parenting cluster characteristics can be used for labelling
and identifying more general processes in the atmosphere, while the lower level clusters can provide information about more
specific properties and features of the observations.
The centroids of the clusters represent characteristics of the points belonging to a cluster in the multivariate input space (in
this case, the polarimetric radar variables and temperature). The identification of these centroids allows the clusters to be
tracked in time and altitude as the centroids are calculated based on QVP from single scans. An analysis of the time series of
the radar volume scans is possible and would allow the clusters to be tracked in time and 3D space. Though unexplored in this

study, the application of the presented approach in this way could be used to provide information on the temporal evolution of the identified hydrometeors and reveal relationships between the identified classes.

Note that the final set of clusters is optimal only for the provided input dataset (Table 1), which gives the user an opportunity to select the input dataset depending on their needs. Thus, for the clustering to reflect ice properties and processes, the appropriate input data climatology should be used. For identification of specific features in the data (e.g. birds or insects) a subset of cases potentially including these features should be selected for the analysis. Further analysis of long-term dataset could be used to create a set of climatologically representative clusters that could be used to study general processes and inform the development of an operational HC scheme.

In this paper, the technique was used for classification of QVP of long-lasting precipitation events, but the same algorithm can be applied to various needs (e.g. identification of birds, insects' or clustering of volume scans of radar data). In parallel with the application of hierarchical clustering technique to other radar observations a thorough validation of the clusters using CVPs following the FAAM BAe-146 is planned.

**Acknowledgments**

The authors would like to thank NCAS for support of all scientific activities and, in particular, the FAAM BAe-146 team for collecting the in situ observations. Airborne data was obtained using the BAe-146-301 Atmospheric Research Aircraft [ARA] flown by Airtask Ltd and managed by FAAM Airborne Laboratory, jointly operated by UKRI and the University of Leeds. The authors are grateful to Hannah Price, Chris Reed and Graeme Nott for their help with CIP data processing. This work was supported by the Natural Environment Research Council [grant numbers NE/P012426/1, NE/S001298/1] and used JASMIN, the UK collaborative data analysis facility.

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

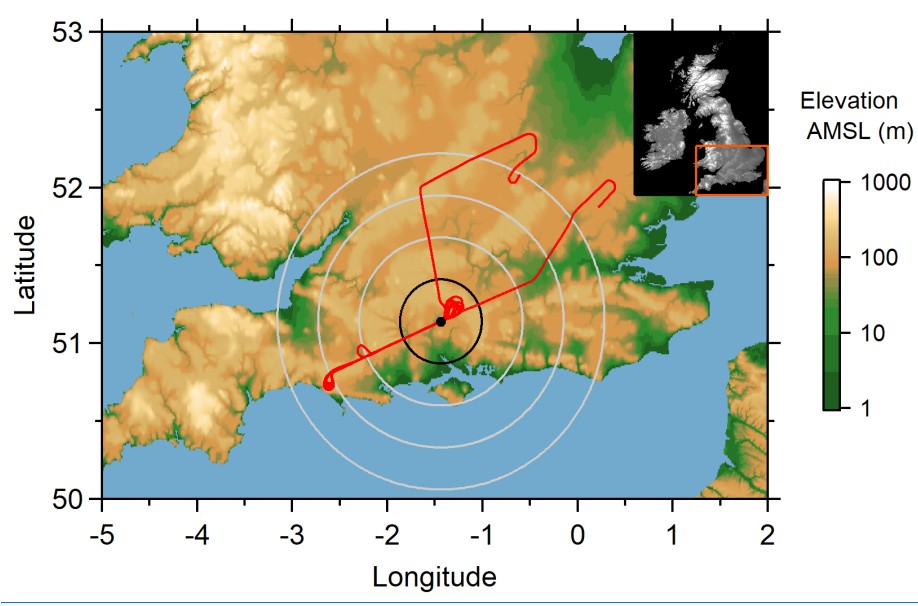

**Figure 1:** NXPol radar location at the Chilbolton Atmospheric Observatory. Circles with the centre at the radar position represent 30, 60, 90 and 120 km range. Credit: USGS (2006).

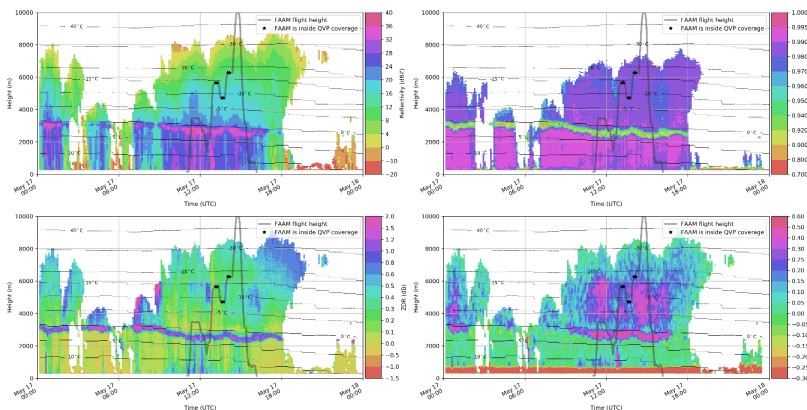

3 **Figure 2:** The time-vs-height QVP of $Z_H$ [dBZ], $Z_{DR}$ [dB], $\rho_{HV}$ [–], and $K_{DP}$ [° km$^{-1}$] retrieved from the NXPol radar observations at

4 Chilbolton on 2017-05-17. Overlaid by temperature isotherms $T$ [° C].

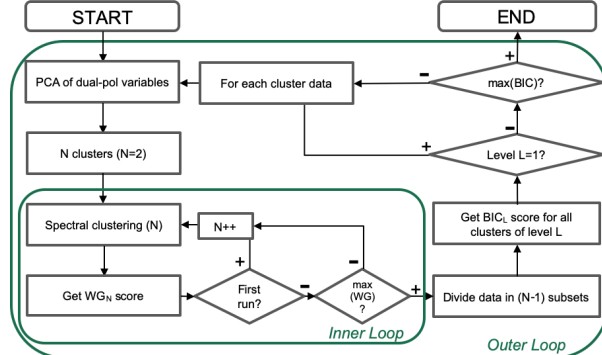

7 **Figure 3:** Flow chart of the implemented hierarchical top-down clustering algorithm.

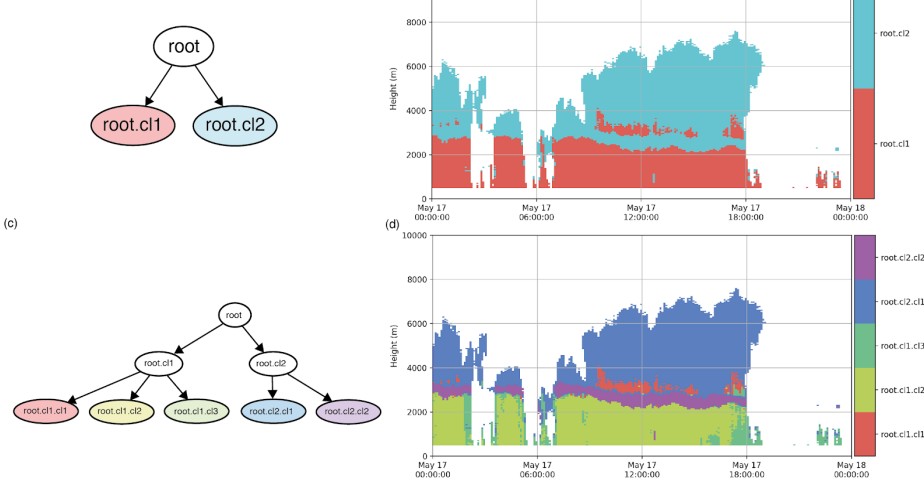

**Figure 4:** The active clusters at the end of the first (panels (a) and (b)) and second (panels (c) and (d)) cycle of the outer loop of the
hierarchical clustering algorithm. Panels (a) and (c) plotted in the hierarchical (tree) structure and panels (b) and (d) plotted in time-vs-
height format of the observations on the 17 of May 2017.

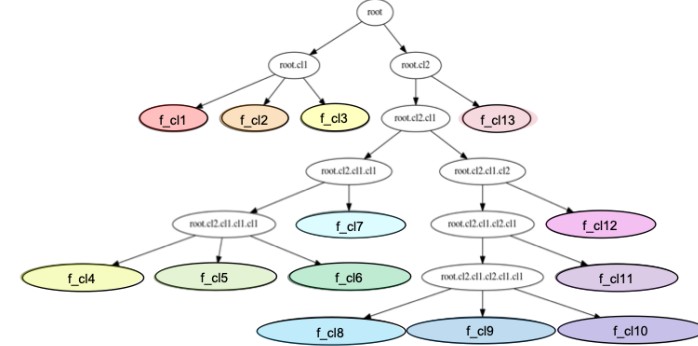

**Figure 5:** Final hierarchical structure of the optimal clustering found for the QVP input data described in Table 1. The final set of optimal
clusters consists of coloured clusters f_cl1, f_cl2, …, f_cl13.

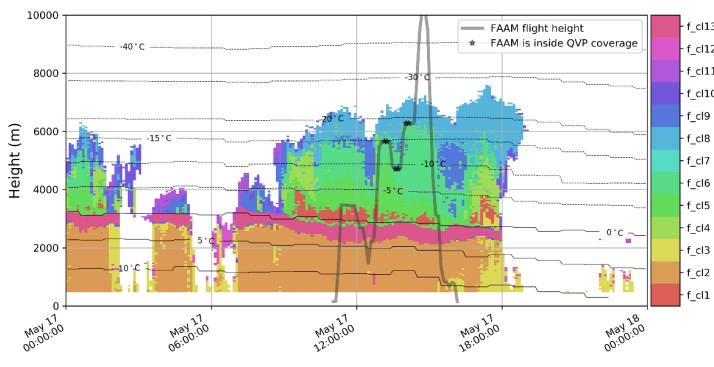

**Figure 6:** The time-vs-height format of the final optimal set of active clusters found by the top-down hierarchical clustering for the QVP
input data described in Table 1 (The hierarchical structure behind the optimal clustering is found in Fig. 4). Example of clusters in the time-
vs-height format of the 17 of May 2017 QVP presented in Fig. 2.

**Figure 7:** Characteristics of the optimal clustering centroids in four polarimetric variables and temperature. The scales of the variables: from -20 dBZ to 40 dBZ for $Z_H$, from -1.5 dB to 2.0 dB for $Z_{DR}$, from 0.9 to 1.0 in $\rho_{HV}$, from -0.3 ° km$^{-1}$ to 0.6 ° km$^{-1}$ for $K_{DP}$, and from -20° C to 10° C in temperature (T).

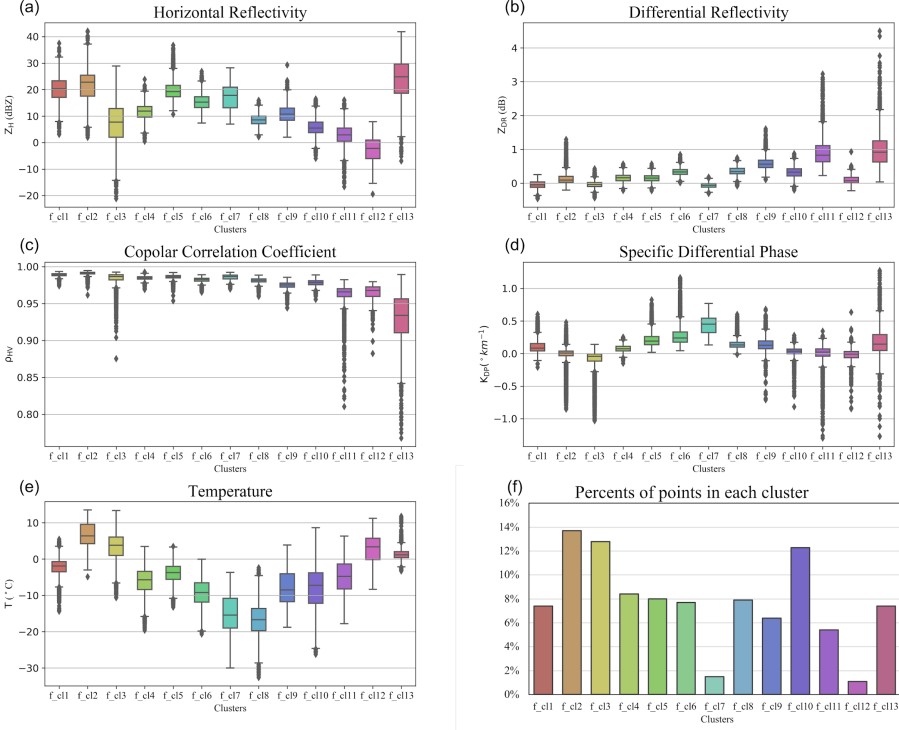

Figure 8: Characteristics of the optimal clustering centroids in four polarimetric variables ((a) – $Z_H$, (b) - $Z_{DR}$, (c) – $\rho_{HV}$, (d) – $K_{DP}$) and temperature (e). The percentage of points in each cluster is in the panel (f).

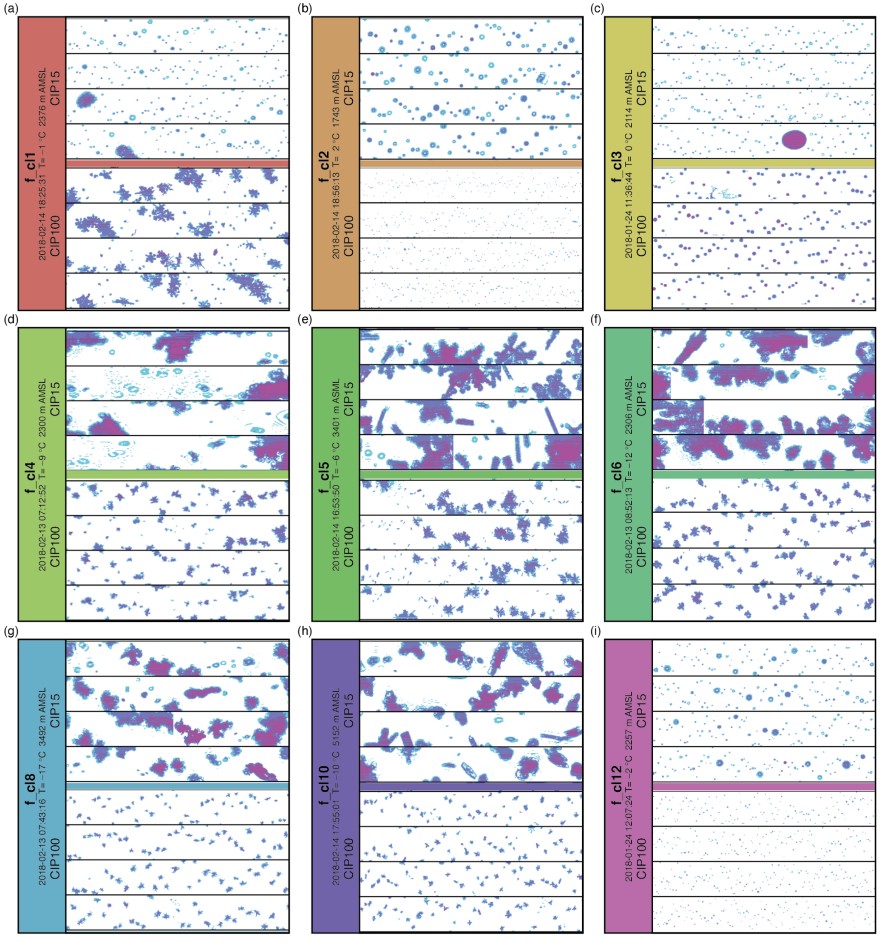

**Figure 9:** Examples of the images taken from the Cloud Image Probe CIP-15 (15$\mu$m, upper) and CIP-100 (100$\mu$m, lower) within 30 km
range from the radar position: (a) for f_cl1 – on 18:25:31 UTC 14-02-2018; (b) for f_cl2 – on 18:56:13 UTC 14-02-2018; (c) f_cl3 – on
11:36:44 UTC 24-01-2018; (d) f_cl4 – on 18:17:53 UTC 14-02-2-18; (e) f_cl5 – on 16:53:50 UTC 14-02-2018; (f) f_cl6 – on 08:52:13 UTC
13-02-2018; (g) f_cl8 – on 07:43:16 UTC 13-02-2018; (h) f_cl10 – on 17:55:01 UTC 14-02-2018; (i) f_cl12 – on 12:07:24 UTC 24-01-
2018.The image widths are 960 and 6400 µm, respectively. The temperature values are derived from the model data and the heights are
derived from the location of the clusters in the QVP.

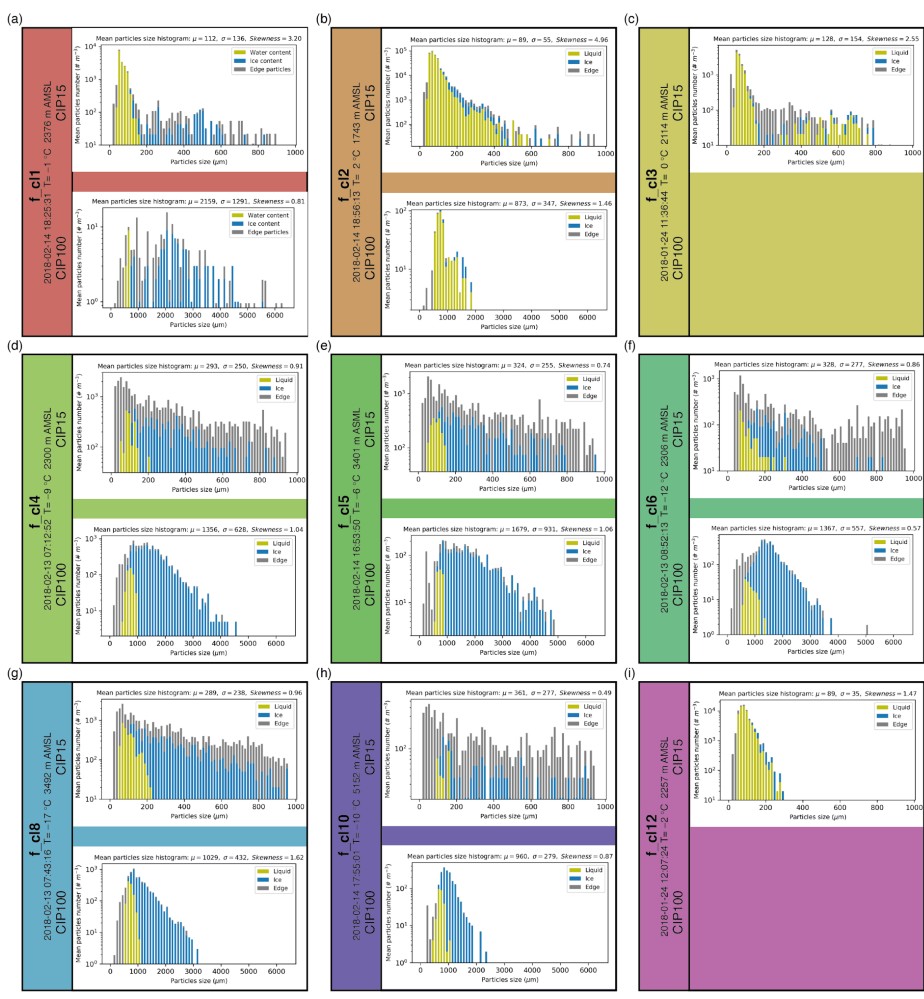

Figure 10: Corresponding to Fig. 9 particle size distributions from the Cloud Image Probe CIP-15 (15$\mu$m, upper) and CIP-100 (100$\mu$m, lower) within 30 km range from the radar position: (a) for f_cl1 – on 18:25:31 UTC 14-02-2018; (b) for f_cl2 – on 18:56:13 UTC 14-02-2018; (c) f_cl3 – on 11:36:44 UTC 24-01-2018; (d) f_cl4 – on 18:17:53 UTC 14-02-2-18; (e) f_cl5 – on 16:53:50 UTC 14-02-2018; (f) f_cl6 – on 08:52:13 UTC 13-02-2018; (g) f_cl8 – on 07:43:16 UTC 13-02-2018; (h) f_cl10 – on 17:55:01 UTC 14-02-2018; (i) f_cl12 – on

12:07:24 UTC 24-01-2018.The image widths are 960 and 6400 μm, respectively. The temperature values are derived from the model data and the heights are derived from the location of the clusters in the QVP.

**Table 1: in situ data collection campaigns**

| Date | FAAM flight number | Number of volume scans | Number of QVP voxels |
|---|---|---|---|
| 20170201 | - | 243 | 46656 |
| 20170203 | - | 213 | 40896 |
| 20170303 | - | 213 | 40896 |
| 20170322 | - | 213 | 40896 |
| 20170517 | C013 | 100 | 19226 |
| 20180124 | C076 | 196 | 37632 |
| 20180213 | C081 | 189 | 36288 |
| 20180214 | C082 | 189 | 36288 |

**Appendix A: Polarimetric characteristics of the optimal clusters**

Table A1 provides the relevant statistics of each of the thirteen optimal clusters identified in this work from a database of X-band radar data.

**Table A1.** Statistics describing the content of the thirteen optimal clusters identified in Sects. 4. For each polarimetric variable and for each cluster, we provide the mean value, standard deviation σ, and [minimum, maximum] values.

| Variable | Cluster | Unit | MeanValue | σ | MinValue | 25 % | 50 % | 75 % | MaxValue | Label |
|---|---|---|---|---|---|---|---|---|---|---|
| $Z_H$ | f_cl1 | dBZ | 20 | 5 | 3 | 17 | 20 | 23 | 38 | aggregation/ riming of ice crtstals |
| $Z_{DR}$ | | dB | -0.04 | 0.11 | -0.46 | -0.12 | -0.05 | 0.03 | 0.25 | |
| $\rho_{HV}$ | | - | 0.989 | 0.002 | 0.974 | 0.987 | 0.989 | 0.990 | 0.994 | |
| $K_{DP}$ | | ° km$^{-1}$ | 0.1 | 0.1 | -0.2 | 0.0 | 0.1 | 0.2 | 0.6 | |
| T | | ° C | -2 | 3 | -14 | -4 | -2 | -1 | 5 | |

| | | | | | | | | | | |
|---|---|---|---|---|---|---|---|---|---|---|
| $Z_H$ | f_cl2 | dBZ | 22 | 6 | 2 | 18 | 23 | 25 | 42 | rain |
| | | | | | | | | | | |
| $Z_{DR}$ | | dB | 0.13 | 0.17 | -0.21 | 0.02 | 0.08 | 0.20 | 1.29 | |
| $\rho_{HV}$ | | - | 0.991 | 0.002 | 0.961 | 0.990 | 0.991 | 0.992 | 0.995 | |
| $K_{DP}$ | | $^\circ\,km^{-1}$ | 0.0 | 0.2 | -0.9 | 0.0 | 0.0 | 0.0 | 0.5 | |
| T | | $^\circ$ C | 7 | 3 | -5 | 4 | 6 | 10 | 14 | |
| $Z_H$ | f_cl3 | dBZ | 7 | 8 | -21 | 2 | 8 | 13 | 29 | light rain/drizzle |
| $Z_{DR}$ | | dB | -0.04 | 0.09 | -0.43 | -0.10 | -0.04 | 0.02 | 0.43 | |
| $\rho_{HV}$ | | - | 0.984 | 0.008 | 0.875 | 0.982 | 0.987 | 0.989 | 0.993 | |
| $K_{DP}$ | | $^\circ\,km^{-1}$ | -0.1 | 0.2 | -1.0 | -0.1 | 0.0 | 0.0 | 0.1 | |
| T | | $^\circ$ C | 4 | 4 | -10 | 1 | 4 | 6 | 13 | |
| $Z_H$ | f_cl4 | dBZ | 12 | 3 | 1 | 10 | 12 | 14 | 24 | low concentration pristine ice & aggregates |
| $Z_{DR}$ | | dB | 0.15 | 0.11 | -0.22 | 0.07 | 0.16 | 0.23 | 0.57 | |
| $\rho_{HV}$ | | - | 0.984 | 0.002 | 0.969 | 0.983 | 0.985 | 0.986 | 0.993 | |
| $K_{DP}$ | | $^\circ\,km^{-1}$ | 0.1 | 0.1 | -0.15 | 0.04 | 0.07 | 0.11 | 0.25 | |
| T | | $^\circ$ C | -6 | 4 | -20 | -8 | -6 | -3 | 3 | |
| $Z_H$ | f_cl5 | dBZ | 20 | 3 | 11 | 17 | 19 | 22 | 37 | low concentration larger aggregates |
| $Z_{DR}$ | | dB | 0.15 | 0.11 | -0.22 | 0.07 | 0.15 | 0.22 | 0.57 | |
| $\rho_{HV}$ | | - | 0.986 | 0.003 | 0.954 | 0.985 | 0.986 | 0.988 | 0.992 | |
| $K_{DP}$ | | $^\circ\,km^{-1}$ | 0.2 | 0.1 | 0.0 | 0.1 | 0.2 | 0.3 | 0.8 | |
| T | | $^\circ$ C | -4 | 3 | -13 | -6 | -4 | -2 | 3 | |
| $Z_H$ | f_cl6 | dBZ | 15 | 3 | 7 | 13 | 15 | 17 | 27 | higher concentration dendritic |
| $Z_{DR}$ | | dB | 0.33 | 0.11 | 0.01 | 0.26 | 0.33 | 0.40 | 0.84 | |

| | | | | | | | | | | |
|---|---|---|---|---|---|---|---|---|---|---|
| $\rho_{HV}$ | | - | 0.982 | 0.003 | 0.965 | 0.980 | 0.983 | 0.984 | 0.989 | crystals & low concentration aggregates |
| $K_{DP}$ | | $°\,km^{-1}$ | 0.3 | 0.2 | 0.0 | 0.2 | 0.2 | 0.3 | 1.2 | |
| T | | $°C$ | -9 | 4 | -21 | -12 | -9 | -7 | 0 | |
| $Z_H$ | f_cl7 | dBZ | 17 | 4 | 7 | 13 | 18 | 21 | 28 | high concentration pristine ice & low concentration larger aggregates |
| $Z_{DR}$ | | dB | -0.06 | 0.08 | -0.03 | -0.12 | -0.06 | -0.01 | 0.18 | |
| $\rho_{HV}$ | | - | 0.985 | 0.004 | 0.970 | 0.984 | 0.987 | 0.988 | 0.992 | |
| $K_{DP}$ | | $°\,km^{-1}$ | 0.4 | 0.1 | 0.1 | 0.3 | 0.5 | 0.5 | 0.8 | |
| T | | $°C$ | -15 | 6 | -30 | -20 | -15 | -10 | -4 | |
| $Z_H$ | f_cl8 | dBZ | 9 | 2 | 2 | 7 | 9 | 10 | 15 | high concentration pristine ice & low concentration of dendrites |
| $Z_{DR}$ | | dB | 0.35 | 0.13 | -0.08 | 0.28 | 0.34 | 0.44 | 0.75 | |
| $\rho_{HV}$ | | - | 0.981 | 0.003 | 0.960 | 0.979 | 0.982 | 0.984 | 0.989 | |
| $K_{DP}$ | | $°\,km^{-1}$ | 0.1 | 0.1 | 0.0 | 0.1 | 0.1 | 0.2 | 0.6 | |
| T | | $°C$ | -17 | 4 | -33 | -20 | -17 | -14 | -2 | |
| $Z_H$ | f_cl9 | dBZ | 11 | 33 | 2 | 8 | 11 | 13 | 29 | dry aggregates of pristine ice |
| $Z_{DR}$ | | dB | 0.59 | 0.18 | 0.10 | 0.46 | 0.56 | 0.68 | 1.61 | |
| $\rho_{HV}$ | | - | 0.975 | 0.004 | 0.944 | 0.972 | 0.975 | 0.977 | 0.985 | |
| $K_{DP}$ | | $°\,km^{-1}$ | 0.1 | 0.1 | -0.7 | 0.1 | 0.1 | 0.2 | 0.7 | |
| T | | $°C$ | -8 | 5 | -19 | -12 | -8 | -4 | 4 | |
| $Z_H$ | f_cl10 | dBZ | 6 | 3 | -6 | 4 | 5 | 8 | 17 | low concentration of pristine ice & dendrites |
| $Z_{DR}$ | | dB | 0.32 | 0.16 | -0.21 | 0.22 | 0.32 | 0.42 | 0.87 | |
| $\rho_{HV}$ | | - | 0.978 | 0.004 | 0.956 | 0.976 | 0.9979 | 0.981 | 0.989 | |
| $K_{DP}$ | | $°\,km^{-1}$ | 0.0 | 0.0 | -0.8 | 0.0 | 0.0 | 0.1 | 0.3 | |
| T | | $°C$ | -8 | 5 | -26 | -12 | -7 | -4 | 9 | |

| | | | | | | | | | | |
|---|---|---|---|---|---|---|---|---|---|---|
| $Z_H$ | f_cl11 | dBZ | 3 | 4 | -17 | 1 | 3 | 5 | 16 | pristine ice crystals |
| $Z_{DR}$ | | dB | 0.92 | 0.43 | 0.23 | 0.63 | 0.83 | 1.11 | 3.23 | |
| $\rho_{HV}$ | | - | 0.961 | 0.017 | 0.810 | 0.959 | 0.966 | 0.970 | 0.982 | |
| $K_{DP}$ | | $°\,km^{-1}$ | 0.0 | 0.2 | -1.3 | 0.0 | 0.0 | 0.1 | 0.3 | |
| T | | $°C$ | -5 | 4 | -18 | -8 | -5 | -1 | 6 | |
| $Z_H$ | f_cl12 | dBZ | -3 | 5 | -19 | -6 | -2 | 1 | 8 | drizzle |
| $Z_{DR}$ | | dB | 0.10 | 0.13 | -0.23 | 0.01 | 0.07 | 0.17 | 0.93 | |
| $\rho_{HV}$ | | - | 0.097 | 0.012 | 0.88 | 0.960 | 0.968 | 0.973 | 0.980 | |
| $K_{DP}$ | | $°\,km^{-1}$ | 0.0 | 0.1 | -0.8 | -0.1 | 0.0 | 0.0 | 0.6 | |
| T | | $°C$ | 3 | 4 | -8 | 0 | 3 | 6 | 11 | |
| $Z_H$ | f_cl13 | dBZ | 24 | 8 | -7 | 19 | 25 | 30 | 42 | MP |
| $Z_{DR}$ | | dB | 0.99 | 0.49 | 0.04 | 0.63 | 0.92 | 0.96 | 0.99 | |
| $\rho_{HV}$ | | - | 0.931 | 0.032 | 0.768 | 0.910 | 0.934 | 0.956 | 0.989 | |
| $K_{DP}$ | | $°\,km^{-1}$ | 0.2 | 0.2 | -1.3 | 0.0 | 0.2 | 0.3 | 1.3 | |
| T | | $°C$ | 1 | 2 | -3 | 0 | 1 | 2 | 11 | |

**Appendix B: The optimal clusters in eight events**
**Figure B1:** The time-vs-height format of the final optimal set of active clusters found by the top-down hierarchical clustering for the QVP
input data described in Table 1. The observations were made on (a) 2017-02-01, (b) 2017- 02-03, (c) 2017-03-03, (d) 2017-03-22, (e) 2017-
05-17, (f) 2018-01-24, (g) 2018-02-13, and (h) 2018-02-14. (The hierarchical structure behind the optimal clustering is found in Fig. 4).