# Peer review of "Hydrometeor classification of quasi-vertical profiles of polarimetric"

_Atmospheric Measurement Techniques, 2020_

## Referee Comment (RC1) · Anonymous Referee #1 · 11 Jun 2020

June 11, 2020

**Overview**

The manuscript presents an approach to hydrometeor classification based on clustering. It contributes therefore to the relatively new family of algorithms and approaches that are built on collected data rather than on simulations (Grazioli et al., 2014a; Bechini and Chandrasekar, 2015; Besic et al., 2016, 2018)

The topic is certainly of interest for the readership of this journal, and I see two main contributions (with respect to previous research) of this manuscript:

- A better conceived usage of clustering, in a smartly designed algorithm.
- The method is designed for QVP data, which are becoming very interesting in terms of microphysical interpretation and noise removal.

I have several comments and suggestions for the authors, to be answered before final acceptance. I recommend overall a major revision.

**Major aspects**

 I found the weakest aspect of this manuscript to be how the particle imagers have been used to provide a physical interpretation of the content of the clusters. This is still largely done by visual inspection, while classification techniques have been developed in the past years to automatically classify hydrometeors from various imagers.

Additionally, I am a bit concerned that the maximum size of particles observable by the airborne instruments is small with respect to the sizes of interest for weather radars. The hydrometeor type of the largest particles is especially important because they often dominate the  $Z_H$  or  $Z_{DR}$  radar signatures (while there are less concerns on  $K_{dp}$  in this sense).

Finally, Section 5.3 is quite hard to follow because of the large use of interpretation and because the clusters are still named by their "anonymous label" (f\_c14...), which makes the narration very dense. Could it be split into smaller sections?

- 2. There seem to be only a few precipitation events contributing to the dataset. As the authors acknowledge in the conclusions, the classification will then be representative only of this dataset. Is it possible to extend the dataset of this research, and try to achieve a hydrometeor classification that is representative of all the weather types that can be expected (given your radar and your geographical location, of course)?
- 3. I strongly recommend to make available (on github or other platform) the codes and some sample of data. This aspect is becoming crucial in modern research, and it will give significant visibility to this work.
4. The authors should better underline, in my opinion, the difference (a difference that affects also the interpretation of polarimetric variables) between hydrometeor classification applied to radar scans, and hydrometeor classification applied to QVPs. I think it is a key novelty of the manuscript.

**Detailed comments**

I refer here to the line numbers of the .PDF version of the manuscript, although I suspect that the numbers are partially cut by the left margin of the pages.

- 1. P2, L41-43: I would recommend the additional effort to specify which of the cited works are based on fuzzy logic, which are based on clustering, and which ones are based on neural networks.
- 2. P2, L46: A detail. I believe that  $Z_H$  should be indicated as "horizontal reflectivity factor" (and the authors may provide the units as [dBZ]). The expression "log-arithmic reflectivity factor" sounds to me not common in the radar meteorology jargon.
- 3. P2, L47: I believe that  $\Phi_{DP}$  should be the "differential phase shift on propagation", to differentiate it from the "total differential phase shift" ( $\Psi_{dp}$ ) which includes also the backscattering contribution. See for example Otto and Russchenberg (2011); Grazioli et al. (2014b).
- 4. P2/3: To complete your review of the state of the art of hydrometeor classification, I recommend also Besic et al. (2018) which tackles the problem of hydrometeor mixtures, and the very smart spatial approach to classification through clustering proposed by Bechini and Chandrasekar (2015). The latter at least partially considers the temporal dimension.
- 5. P4, L89: the usage of PCA is quite common and always tempting. However, which are the underline assumptions on the input data when we use PCA on them?
- 6. P5, L51: why those two evaluation scores have been chosen among the many available in the literature? In my experience, this is really a key point and often the answer is very different according to different evaluation scores.
- 7. P5: I would recommend to provide a few "qualitative" words about the indexes used. For example, if I interpret well, WG is an index that values compactness and separation. It could be stated. A visual aid, even with simple 2D data, would help the readers who are less familiar with those techniques.
- 8. P6, L78: a "modified" Meteor 50DX. The authors may want to specify the modification (i believe for this radar is a bigger antenna dish, but I may be wrong).
- 9. P7, L84:  $K_{dp}$  is a derived quantity. Could you please mention which estimation method was used?
- 10. P7: how are  $Z_H$  and  $Z_{DR}$  corrected for signal attenuation?
- 11. P7, L96-98: in more complex terrains (Alps for example), I believe that the errors of such an approach may be larger.
- 12. P8, L31: for the hydrometeor sizes of interest for X-band weather radars, isn't  $6350\mu$ m still small?
- 13. P9, L68: About data standardization. Is it a simple standardization based on mean and standard deviation before to apply PCA, do I interpret it correctly? Are very skewed variables,  $K_{dp}$  for instance, treated differently?
- 14. P10: it is not clear to me, maybe I missed it, how much data is used as input of the clustering. Is it representative of various seasons? How is it chosen? The
clusters will then be representative of this dataset, so it is important to clarify this point.

- 15. P12, L38: I suspect that part of the variation of  $K_{dp}$  among clusters is due to the fact that it is a variable with an extremely skewed typical distribution.
- P14/15: I could recommend, if they can help the discussion and the interpretation, the following researches dedicated to ice-phase microphysics: Bechini et al. (2013); Grazioli et al. (2015); Kennedy and Rutledge (2011)
- 17. P16, L82/83: considering past papers on the topic, I found slightly overstated to claim this research to be the first tackling the issue of the number of clusters as they appear in the data. The authors may consider to specify more in detail the novel aspects of their approach here.
- 18. Figure 7: a very nice way to display the clusters, although it is a bit hard to see the different types of lines, corresponding to the different days.

**References**

- J. Grazioli, D. Tuia, S. Monhart, M. Schneebeli, T. Raupach, and A. Berne. Hydrometeor classification from two-dimensional video disdrometer data. *Atmos. Meas. Tech.*, 7(9):2869–2882, 2014a. doi: 10.5194/amt-7-2869-2014.
- R. Bechini and V. Chandrasekar. A semisupervised robust hydrometeor classification method for dual-polarization radar applications. *J. Atmos. Oceanic Technol.*, 32(1):22–47, 2015. doi: 10.1175/JTECH-D-14-00097.1.
- N. Besic, J. Figueras i Ventura, J. Grazioli, M. Gabella, U. Germann, and A. Berne. Hydrometeor classification through statistical clustering of polarimetric radar measurements: a semi-supervised approach. *Atmos. Meas. Tech.*, 9(9):4425–4445, 2016. doi: 10.5194/ amt-9-4425-2016. URL https://www.atmos-meas-tech.net/9/4425/2016/.

AMTD
- N. Besic, J. Gehring, C. Praz, J. Figueras i Ventura, J. Grazioli, M. Gabella, U. Germann, and A. Berne. Unraveling hydrometeor mixtures in polarimetric radar measurements. *Atmos. Meas. Tech.*, 11(8):4847–4866, AUG 22 2018. doi: 10.5194/amt-11-4847-2018.
- T. Otto and H. W. J. Russchenberg. Estimation of specific differential phase and differential backscatter phase from polarimetric weather radar measurements of rain. *IEEE Geosci. Remote Sens. Lett.*, 8(5):988–992, 2011. doi: 10.1109/LGRS.2011.2145354.
- J. Grazioli, M. Schneebeli, and A. Berne. Accuracy of phase-based algorithms for the estimation of the specific differential phase shift using simulated polarimetric weather radar data. *IEEE Geosci. Remote Sens. Lett.*, 11(4):763–767, 2014b. doi: 10.1109/LGRS.2013.2278620.
- R. Bechini, L. Baldini, and V. Chandrasekar. Polarimetric radar observations in the ice region of precipitating clouds at c-band and x-band radar frequencies. J. Appl. Meteor. Clim., 52: 1147–1169, 2013. doi: 10.1175/JAMC-D-12-055.1.
- J. Grazioli, G. Lloyd, L. Panziera, C. R. Hoyle, P. J. Connolly, J. Henneberger, and A. Berne. Polarimetric radar and in situ observations of riming and snowfall microphysics during clace 2014. Atmos. Chem. Phys., 15(23):13787–13802, 2015. doi: 10.5194/acp-15-13787-2015. URL https://www.atmos-chem-phys.net/15/13787/2015/.
- P. C. Kennedy and S. A. Rutledge. S-band dual polarization radar observations of winter storms. *J. Appl. Meteor. Clim.*, 50(4), 2011. doi: 10.1175/2010JAMC2558.1.

---

## Referee Comment (RC2) · Anonymous Referee #2 · 23 Jun 2020

The paper presents a new approach for unsupervised hydrometeor classification using polarimetric radar data and temperature. The suggested procedure uses a hierarchical clustering methodology to determine a number of data clusters that can be objectively distinguished from the multiparameter observations. These clusters can be associated with certain classes of hydrometeors using either in situ observations or general physical considerations. It is important that the study utilizes the quasi-vertical profiles (QVPs) of polarimetric radar variables which capture the vertical structure of the cloud and its temporal evolution with high vertical resolution and statistical accuracy. Another positive aspect of this investigation is the utilization of the airborne microphysical probes to label the identified multiparameter data clusters to characterize hydrometeors with different microphysical properties. The paper is well written and is worth to be published after several issues with the analysis and interpretation of the results are addressed. Here are my comments and recommendations (1) The authors do not consider the impact of the measurement errors (biases and statistical errors) on the outcome of their classification. There is little doubt that if the measurements are too noisy then the number of objectively distinguished clusters of data is reduced. The potential biases of ZDR and KDP due to miscalibration of ZDR, differential attenuation, and backscatter differential phase in the melting layer (ML) are not mentioned in the manuscript. The evidence of such biases is indicated by negative values of mean ZDR and KDP for several classes. I strongly recommend at least some discussion about the effects of the measurements errors on the classification results. (2) I do not agree with microphysical labeling of several identified clusters. For example, the cluster f_cl1 is labeled as upper part of ML. However, it is obvious from Table A1 and Figs. 6 and 8 that the corresponding signature is observed at negative temperature above the ML and is likely associated with heavily aggregated or rimed snow. Its melting often produces the sagging of the ML as demonstrated in Fig. 6. Cluster f_cl7 has highest KDP which is a manifestation of the rapid growth of ice via vapor deposition and onset of aggregation in the dendritic growth layer (DGL) centered at -15°C. However, the authors label f_cl9 as DGL although the corresponding temperature is higher than -10°C. DGL is not a hydrometeor class but a layer where dendrites or hexagonal plates typically grow. Depending on the height of the top of the cloud, DGL can be characterized either by a combination of high KDP and low ZDR or low KDP and high ZDR (Griffin et al. 2018, JAMC, pp. 31 − 50). This important characterization is completely missed in the manuscript as well as the reference to the very pertinent Griffin et al. (2018) article. Cluster f_cl12 is labeled as "ambiguous small ice / drizzle". Small ice is very unlikely because it would completely melt at T = 3°C. (3) The onset of melting is determined by a zero value of the wet bulb temperature rather than regular temperature and

I suggest using the wet bulb temperature for classification. (4) For a future studies I would recommend using vertical gradient of Z and the height of the top of the cloud as additional classification variables. Vertical gradient of Z better characterizes the aggregation / riming process than the absolute value of Z. It can be seen from the results of classification presented in the manuscript that the cluster f_cl11 is correctly recognized as pristine ice with low Z and KDP and high ZDR and is identified during time periods when the height of the cloud top was law. Significant aggregation is unlikely in this situation due to low number concentration of ice particles and small difference in their terminal velocities. (5) I notice that several literature references mentioned in a body of the manuscript (Kumjian 2012, Hampton 2019, 2020, Murphy 2018) are missing in the reference list.

---

## Author Comment (AC1) · 20 Oct 2020

**Authors' responses to reviewer #1 comments**

**Title: Hydrometeor classification of quasi-vertical profiles of polarimetric radar measurements using a top-down iterative hierarchical clustering method**

Notes to the reviewer:

The authors would like to thank the reviewer for the feedback and constructive comments on our manuscript. Comments have been taken from the provided review and listed below in numbered list format, and are addressed directly afterword in sub-bullets, for example:

1.   Reviewer Comments

     Authors' Response

     "citation from the revised manuscript"

As the reviewer comments were copied from the original review, the line numbers there will refer to the reviewed manuscript. Line numbers in our responses refer to the revised manuscript.

Figures that are in the manuscript are referenced just with the number, e.g. Fig. 1.

Reviewer #1:

**Overview**

The manuscript presents an approach to hydrometeor classification based on clustering. It contributes therefore to the relatively new family of algorithms and approaches that are built on collected data rather than on simulations (Grazioli et al., 2014a; Bechini and Chandrasekar, 2015; Besic et al., 2016, 2018) The topic is certainly of interest for the readership of this journal, and I see two main contributions (with respect to previous research) of this manuscript:

• A better conceived usage of clustering, in a smartly designed algorithm.

• The method is designed for QVP data, which are becoming very interesting in terms of microphysical interpretation and noise removal.

I have several comments and suggestions for the authors, to be answered before final acceptance. I recommend overall a major revision.

**Major aspects**

1. I found the weakest aspect of this manuscript to be how the particle imagers have been used to provide a physical interpretation of the content of the clusters. This is still largely done by visual inspection, while classification techniques have been developed in the past years to automatically classify hydrometeors from various imagers.

> We appreciate the Reviewers comments about particle image classification. Whilst image classification may be useful to pursue in the future, there is still significant uncertainty and variation in the quality of these algorithms and the usefulness in this study due to limitations of the observations for large precipitation particles. We have added the following text to p.10, L.1 - 6 to clarify:

>> "No attempts have been made to classify particle images, for two key reasons. (1) The larger particles, which have the greatest influence on the polarimetric properties, are poorly sampled by the CIPs. (2) A recent study by O'Shea et al (2020, in review at AMTD) suggests existing procedures to classify particle images using the CIP can lead to inaccurate results due to the effects of diffraction when particles are imaged more than a few mm off the focal plane, which is the most common scenario. A thorough assessment of the accuracy of these image classification algorithms, with respect to particle size and probe configuration, is much needed."

2. Additionally, I am a bit concerned that the maximum size of particles observable by the airborne instruments is small with respect to the sizes of interest for weather radars. The hydrometeor type of the largest particles is especially important because they often dominate the $Z_H$ or $Z_{DR}$ radar signatures (while there are less concerns on $K_{dp}$ in this sense).

> We appreciate the Reviewer's concern about the maximum observable particles in the CIP100 data. We agree that the in-situ measurements are limited in the ability to quantify the concentration of larger particles, which may have an important contribution to the bulk polarimetric properties. We have added the following text p.9, L.13 - 17 to clarify:

>> "It should be noted that despite using the centre-in method with the CIP data, which increases the effective sample volume for larger particles at the expense of uncertainty in particle size, the ability to measure particles with size > 6 mm is negligible with this configuration. An indication of the potential presence of such large particles can be obtained through a visual inspection of the particle images, but no conclusions can be drawn."

3. Finally, Section 5.3 is quite hard to follow because of the large use of interpretation and because the clusters are still named by their "anonymous label" (f_c14. . . ), which makes the narration very dense. Could it be split into smaller sections?

We thank the Reviewer for the suggestion and acknowledge that the section might be difficult to read. We split Section 5.3 into four subsections.

4. There seem to be only a few precipitation events contributing to the dataset. As the authors acknowledge in the conclusions, the classification will then be representative only of this dataset. Is it possible to extend the dataset of this research, and try to achieve a hydrometeor classification that is representative of all the weather types that can be expected (given your radar and your geographical location, of course)?

We appreciate the Reviewer's concern about the representativeness of the dataset. To clarify our view on the purpose of this study we added a paragraph to p. 3, L. 16 - 19:

"The point of this study is not to create a set of cluster characteristics that could be applied to other datasets. Rather the goal is to demonstrate the viability of this type of data-driven methodology for creating a set of labelled clusters (i.e. hydrometeor classes). As such, the comparison to in situ data and labelling done as part of this study is only shown as an example of how this tool can be used. "

5. I strongly recommend to make available (on github or other platform) the codes and some sample of data. This aspect is becoming crucial in modern research, and it will give significant visibility to this work.

We thank the Reviewer for this recommendation. The samples are available for both the radar and the CIP data on the Centre for Environmental Data Analysis (CEDA) archive. The links to the data can be found in the correspondent references (p.8, L.31 - p.9, L.2).

"In situ data for this study comes from FAAM BAe-146 flights C013, C076, C081, and C082 (FAAM, 2017; FAAM, 2018a; FAAM, 2018b; FAAM, 2018c) and observational data are available on CEDA archive. The dates of the flights with corresponding flight numbers can be found in Table 1."

The manuscript is updated by adding:

"Radar data selected for this study can be found in the list of mobile X-band radar observations on CEDA archive (Bennett, 2020)."

and

"The original data can be found on CEDA archive (Met Office, 2016)."

to p. 7, L.16 - 17 and p. 8, L. 5 .

The QVP extraction and the clustering code can be made available on request to the authors and this is mentioned in the new version of the manuscript (p. 10, L. 16).

"and the code can be made available on request."

6. The authors should better underline, in my opinion, the difference (a difference that affects also the interpretation of polarimetric variables) between hydrometeor classification applied to radar scans, and hydrometeor classification applied to QVPs. I think it is a key novelty of the manuscript.

We appreciate the Reviewer's feedback on this. Seeing this comment, we agree that more discussion is warranted. As such we have added

"Note that this is a novel application of the QVP data product and the interpretation of QVP polarimetric variables differs from that of PPI or RHI scans due to the averaging used to construct them."

to p. 3, L. 27 - 29 and

"As QVP polarimetric characteristics differ from polarimetric characteristics of hydrometeors observed by PPI and RHI scans, care must be taken when comparing these results to the literature."

to p. 12, L. 6 - 8.

We also added a discussion on the influence of errors and noise in the data on the clustering and labeling, where QVP advantages are mentioned one more time (p. 14, L. 30 - p. 15, L. 3):

"Measurement errors may influence the clustering results. As it was shown by Bringi et al. (1990) noise in the observations has a strong impact on k-mean HCA results. Unfortunately, it is impossible to run the same type of analysis conducted by Bringi et al. (1990) for a unsupervised hierarchical clustering algorithm as the added noise might deliver modified hierarchical structure with another optimal number of clusters and direct comparison to the original set of final clusters would be impossible. Here this issue of noise is partially addressed through our use of QVPs. In particular, azimuthal averaging of a QVP reduces the noisiness of the differential phase within the melting layer, Trömel et al. (2013, 2014) and was recommended in Kumjian et al. (2013) to quantify rather small enhancements of $Z_{DR}$ and $K_{DP}$."

**Detailed comments**

7. P2, L41-43: I would recommend the additional effort to specify which of the cited works are based on fuzzy logic, which are based on clustering, and which ones are based on neural networks.

> We appreciate the Reviewer's suggestion. The sentence:

>> "Further refinement and development of automatic HC algorithms included the application of fuzzy-logic, machine-learning techniques (such as the identification of clusters representing data-wise similarities) and neural networks (Straka et al., 2000; Liu and Chandrasekar, 2000; Wen et al.,2015; Grazioli et al., 2015; Besic et al., 2016; Wang et al., 2017; and Ribaud et al., 2019)."

> Is now modified to p.2, L. 17 - 20:

>> "Further refinement and development of automatic HC algorithms included the application of fuzzy-logic (Straka et al., 2000; Liu and Chandrasekar, 2000), machine-learning techniques (such as the identification of clusters representing data-wise similarities) (Wen et al.,2015; Grazioli et al., 2015; Besic et al., 2016; and Ribaud et al., 2019) and neural networks (Wang et al., 2017). "

8. P2, L46: A detail. I believe that $Z_H$ should be indicated as "horizontal reflectivity factor" (and the authors may provide the units as [dBZ]). The expression "logarithmic reflectivity factor" sounds to me not common in the radar meteorology jargon.

> We thank the Reviewer for this correction. The variable $Z_H$ is referred to as

>> "horizontal reflectivity factor"

> now. The corresponding adjustment is made in the text (p. 2, L. 23).

9. P2, L47: I believe that $\Phi_{DP}$ should be the "differential phase shift on propagation", to differentiate it from the "total differential phase shift" ($\Psi_{DP}$) which also includes the backscattering contribution. See for example Otto and Russchenberg (2011); Grazioli et al. (2014b).

> We have adjusted the variable introduction. The $\Phi_{DP}$ is referred now as

>> "differential phase shift on propagation"

> (p.2, L. 24).

10. P2/3: To complete your review of the state of the art of hydrometeor classification, I recommend also Besic et al. (2018) which tackles the problem of hydrometeor mixtures, and the very smart spatial approach to classification through clustering proposed by Bechini and Chandrasekar (2015). The latter at least partially considers the temporal dimension?

> We appreciate the reviewer suggestion. The references are added to the text on the p.2, L. 22 and p.3, L. 21 - 22.

11. P4, L89: the usage of PCA is quite common and always tempting. However, which are the underline assumptions on the input data when we use PCA on them?

> We thank the Reviewer for stressing the importance of the underlying PCA assumptions. We have adjusted the text adding description of the data preprocessing, required for the application of PCA. The manuscript p.10, L. 19 - 21 now includes:

>> " The non-parametric transformation based on the quantile function maps the data to a uniform distribution. This standardization helps to deal with outliers and satisfy PCA data assumptions."

12. P5, L51: why those two evaluation scores have been chosen among the many available in the literature? In my experience, this is really a key point and often the answer is very different according to different evaluation scores.

> The text:

>> " The WG index was chosen as best performing according to comparison studies (Niemelä et al., 2018 and Hämäläinen et al., 2017). The BIC is best for the calculation of the posterior probability of a clustering."

> was added to the manuscript (p. 6, L. 8 - 10)

13. P5: I would recommend to provide a few "qualitative" words about the indexes used. For example, if I interpret well, WG is an index that values compactness and separation. It could be stated. A visual aid, even with simple 2D data, would help the readers who are less familiar with those techniques.

> We thank the Reviewer for the recommendation. The manuscript is adapted by adding:

>> " The WG-index is a measure of compactness based on the distances between the points and the barycenters of all clusters."

> to p.6, L. 21 - 22 and

>> "The BIC is an estimate of a function of the posterior probability of a clustering being true, under a certain Bayesian setup, so that a higher BIC in (2) means that a clustering is considered to be more likely to be the optimal clustering."

> to p.6, L. 28 - 29.

14. P6, L78: a "modified" Meteor 50DX. The authors may want to specify the modification (i believe for this radar is a bigger antenna dish, but I may be wrong).

> The reviewer is correct, the modification is a non-standard 2.4m antenna. We have changed the text accordingly (p. 7, L. 11 - 12) to make this clear.

"The NXPol is a mobile Meteor 50DX (Leonardo Germany GmbH) X-band, dual-polarization, Doppler weather radar with a 2.4 m diameter antenna."

15. P7, L84: Kdp is a derived quantity. Could you please mention which estimation method was used?

> We thank the Reviewer for the question. We have added
>
> > "Here $K_{DP}$ is calculated as the linear gradient of differential phase shift, where the phase shift has been filtered to remove non-meteorological targets ($\rho_{HV} > 0.85$) and progressively smoothed using decreasing length averaging windows"
>
> to the text on the p.7, L. 21 - 23.

16. P7: how are ZH and ZDR corrected for signal attenuation?

> Both ZH and ZDR are corrected for attenuation using a single factor linear correction based on the differential phase shift on propagation in liquid precipitation. We have added the following to the text to make this clear
>
> > "and $Z_H$ and $Z_{DR}$ are corrected for attenuation using a linear correction (Bringi et al. 1990)"
>
> (p.7, L. 23 - 24).

17. P7, L96-98: in more complex terrains (Alps for example), I believe that the errors of such an approach may be larger.

> We agree that in more complex terrain an alternative approach may be required for temperature rather than using interpolated model output.

18. P8, L31: for the hydrometeor sizes of interest for X-band weather radars, isn't 6350µm still small?

> See our response to the major aspect comment 1 above.

19. P9, L68: About data standardization. Is it a simple standardization based on mean and standard deviation before to apply PCA, do I interpret it correctly? Are very skewed variables, Kdp for instance, treated differently?

> See our response to the 11-th detailed comment.

20. P10: it is not clear to me, maybe I missed it, how much data is used as input of the clustering. Is it representative of various seasons? How is it chosen? The clusters will then be representative of this dataset, so it is important to clarify this point.

> The data used for clustering is shown in Table 1 and we explain that the clustering is only applicable to this dataset in our discussions. We have added an extra

reference to this table on p.19 L. 3 to aid the reader in considering this at an appropriate point.

21. P12, L38: I suspect that part of the variation of Kdp among clusters is due to the fact that it is a variable with an extremely skewed typical distribution.
    Thank you for this comment we added it to the discussion p. 13, L. 5 - 6:

    "In part this variation may be due to the fact that this variable has an extremely skewed distribution."

22. P14/15: I could recommend, if they can help the discussion and the interpretation, the following researches dedicated to ice-phase microphysics: Bechini et al. (2013); Grazioli et al. (2015); Kennedy and Rutledge (2011)
    We thank the reviewer for these helpful suggestions of additional references. We have cited them where appropriate (p. 14, L. 10 - 11):

    "These characteristics were also recognised as a signature of dendritic crystals in Bechini et al. (2013)."

    and p.3, L. 20, L. 22; p. 4, L. 30; p. 7, L. 25.

23. P16, L82/83: considering past papers on the topic, I found slightly overstated to claim this research to be the first tackling the issue of the number of clusters as they appear in the data. The authors may consider to specify more in detail the novel aspects of their approach here.
    It is corrected by adding (p. 18, L. 10 - 11)

    "in top-down hierarchical"

    to the sentence

    "This represents the first attempt, in top-down hierarchical clustering of weather radar data, to identify the number of clusters based solely on the embedded data characteristics."

24. Figure 7: a very nice way to display the clusters, although it is a bit hard to see the different types of lines, corresponding to the different days.

    We thank the reviewer for this suggestion. We have tried to adjust this figure to improve it further but unfortunately thicker lines make all the plots unreadable.

**References:**

O'Shea, S., Crosier, J., Dorsey, J., Gallagher, L., Schledewitz, W., Bower, K., Schlenczek, O., Borrmann, S., Cotton, R., Westbrook, C., and Ulanowski, Z.: Characterising optical array particle imaging probes: implications for small ice crystal observations, Atmos. Meas. Tech. Discuss., https://doi.org/10.5194/amt-2020-265, in review, 2020.

Niemelä, M., Ayrämo, S. and Kärkkäinen, T.: Comparison of Cluster Validation Indices with Missing Data. ESANN 2018 proceedings, European Symposium on Artificial Neural Networks, Computational Intelligence and Machine Learning. Bruges (Belgium), 25-27 April 2018, i6doc.com publ., ISBN 978-287587047-6. Available from http://www.i6doc.com/en/

Hämäläinen, J., Jauhiainen, S., Kärkkäinen, T.: Comparison of Internal Clustering Validation Indices for Prototype-Based Clustering. Algorithms. 10. 105. 10.3390/a10030105, 2017.

---

## Author Comment (AC2) · 20 Oct 2020

**Authors' responses to reviewer #2 comments**

**Title: Hydrometeor classification of quasi-vertical profiles of polarimetric radar measurements using a top-down iterative hierarchical clustering method**

Notes to the reviewer:

The authors would like to thank the reviewer for the feedback and constructive comments on our manuscript. Comments have been taken from the provided review and listed below in numbered list format, and are addressed directly afterword in sub-bullets, for example:

1.  Reviewer Comments

    Authors' Response

    "citation from the revised manuscript"

As the reviewer comments were copied from the original review, the line numbers there will refer to the reviewed manuscript. Line numbers in our responses refer to the revised manuscript.

Figures that are in the manuscript are referenced just with the number, e.g. Fig. 1.

Reviewer #2:

The paper presents a new approach for unsupervised hydrometeor classification using polarimetric radar data and temperature. The suggested procedure uses a hierarchical clustering methodology to determine a number of data clusters that can be objectively distinguished from the multiparameter observations. These clusters can be associated with certain classes of hydrometeors using either in situ observations or general physical considerations. It is important that the study utilizes the quasi-vertical profiles (QVPs) of polarimetric radar variables which capture the vertical structure of the cloud and its temporal evolution with high vertical resolution and statistical accuracy. An other positive aspect of this investigation is the utilization of the airborne microphysical probes to label the identified multiparameter data clusters to characterize hydrometeors with different microphysical properties. The paper is well written and is worth to be published after several issues with the analysis and interpretation of the results are addressed. Here are my comments and recommendations:

1.  The authors do not consider the impact of the measurement errors (biases and statistical errors) on the outcome of their classification. There is little doubt that if the measurements are too noisy then the number of objectively distinguished clusters of data is reduced. The

potential biases of ZDR and KDP due to miscalibration of ZDR, differential attenuation, and backscatter differential phase in the melting layer (ML) are not mentioned in the manuscript. The evidence of such biases is indicated by negative values of mean ZDR and KDP for several classes. I strongly recommend at least some discussion about the effects of the measurements errors on the classification results.

We thank the Reviewer for the comment. The noise and errors in the data might indeed influence the resulting clustering and we added a discussion of these influences on the clustering and labelling of the clusters in Subsection 5.2, p. 14., L.30 - p. 15, L. 11.

> "Measurement errors may influence the clustering results. As it was shown by Bringi et al. (1990) noise in the observations has a strong impact on k-mean HCA results. Unfortunately, it is impossible to run the same type of analysis conducted by Bringi et al. (1990) for an unsupervised hierarchical clustering algorithm as the added noise might deliver modified hierarchical structure with another optimal number of clusters and direct comparison to the original set of final clusters would be impossible. Here this issue of noise is partially addressed through our use of QVP. In particular, azimuthal averaging of a QVP reduces the noisiness of the differential phase within the melting layer (Trömel et al. (2013, 2014)) and was recommended in Kumjian et al. (2013) to quantify rather small enhancements of $Z_{DR}$ and $K_{DP}$.

> The mean negative values of $Z_{DR}$ and $K_{DP}$ in some clusters (f_cl1, f_cl3 and f_cl7) might point at potential biases due to miscalibration of $Z_{DR}$, differential attenuation, or backscatter differential phase in the melting layer. Biases in the data such as miscalibrations will not impact the clustering process but will impact the labelling as it is based on cluster characteristics. A miscalibration of $Z_{DR}$ can also be excluded as we routinely perform calibration of this variable. $Z_H$ and $Z_{DR}$ are corrected for the attenuation in the data pre-processing. Biases caused by backscatter differential phase in the melting layer (Trömel et al., 2014) have not been removed and are evident in the cluster characteristics. The influence of the backscatter differential phase needs further investigation. As discussed, not all clusters can be labelled with absolute confidence solely based on the cluster's characteristics and in situ observations can help to verify these initial suppositions."

We also added a sentence (p. 7, L. 21 - 24) mentioning the attenuation correction of Z and ZDR:

> "Here $K_{DP}$ is calculated as the linear gradient of differential phase shift, where the phase shift has been filtered to remove non-meteorological targets ($\rho_{HV} > 0.85$) and progressively smoothed using decreasing length averaging windows and $Z_H$ and $Z_{DR}$ are corrected for attenuation using a linear correction (Bringi et al. 1990)."

2. I do not agree with microphysical labeling of several identified clusters. For example, the cluster f_cl1 is labeled as upper part of ML. However, it is obvious from Table A1 and Figs. 6 and 8 that the corresponding signature is observed at negative temperature above the ML and is likely associated with heavily aggregated or rimed snow. Its melting often produces the sagging of the ML as demonstrated in Fig. 6.

> The cluster labelling is clarified in the manuscript p.13, L. 25 - 32:

>> " These characteristics immediately indicate that f_cl13 can be labelled as the "bright band" belonging cluster having mixed-phase (MP) particles.

>> The MP cluster (f_cl13 in Fig. 6) is observed to have some sagging areas: between 10:00 (UTC) and 12:20 (UTC), around 16:00 (UTC) and near 18:00 (UTC). Note that f_cl1 is observed above the MP cluster f_cl13 exactly at these time intervals (Fig. 6). This sagging "bright band" signature is often observed where aggregation and riming processes are occurring directly above the melting layer (Kumjian et al., 2016 and Ryzhkov & Zrnic, 2019). This suggests that f_cl1 can be associated with the processes of aggregation or riming and labelled accordingly."

3. Cluster f_cl7 has highest KDP which is a manifestation of the rapid growth of ice via vapor deposition and onset of aggregation in the dendritic growth layer (DGL) centered at -15 °C. However, the authors label f_cl9 as DGL although the corresponding temperature is higher than -10 °C. DGL is not a hydrometeor class but a layer where dendrites or hexagonal plates typically grow. Depending on the height of the top of the cloud, DGL can be characterized either by a combination of high KDP and low ZDR or low KDP and high ZDR (Griffin et al. 2018, JAMC, pp. 31 – 50). This important characterization is completely missed in the manuscript as well as the reference to the very pertinent Griffin et al. (2018) article.

> The description of cluster f_cl7 is modified and a reference to Griffin et al. (2018) is added to the text (p. 14, L. 6-10):

>> "The combination of rather high $Z_H$ (17 dBZ) and high $K_{DP}$ at temperatures around -15 °C indicates a cluster with high particle number concentration of small ice crystals mixed with a small amount of bigger aggregates. This cluster is potentially a manifestation of the rapid growth of ice via vapor deposition and onset of aggregation in the dendritic growth layer (DGL) discussed in the details in Griffin et al. (2018)."

> We agree that DGL is not a hydrometeor class, as well as the melting layer (ML). The referencing to these areas in the data as DGL are deleted in the manuscript (p. 13 - 14, p. 32, and p. 34 - 35).

4. Cluster f_cl12 is labeled as "ambiguous small ice / drizzle". Small ice is very unlikely because it would completely melt at T = 3 °C.

> We appreciate the Reviewer's comment on the assignment of this cluster. The mean 3 °C temperature calculated for the centroid of the cluster would definitely

mean small ice's melting, but if we look at f_cl12 in Fig. 7 we observe the variation of the centroids of cluster data subsets belonging to different dates. For some dates the centroid of this cluster shows a temperature values of -2 °C or even -4 °C and small ice is very possible at these temperatures. Even so, we recognise the median values that describe the cluster characteristics are indicative of what is recognised as drizzle, so we have changed this in the text (p.14, L. 11 - 12):

> " Combining the low mean $Z_H$ with low mean $Z_{DR}$ (0.097 dB) and temperature about 3 °C we can assume that f_cl12 can be labelled as small droplets (i.e. drizzle)."

and in Table A1 p.34.

5.  The onset of melting is determined by a zero value of the wet bulb temperature rather than regular temperature and I suggest using the wet bulb temperature for classification.

    We agree that wet bulb temperature would suit better the purposes of the study but unfortunately the data were not available due to technical limitations. We are planning to work on it in future and will use wet bulb temperature data for the next, more detailed analysis.

6.  For a future studies I would recommend using vertical gradient of Z and the height of the top of the cloud as additional classification variables. Vertical gradient of Z better characterizes the aggregation/ riming process than the absolute value of Z. It can be seen from the results of classification presented in the manuscript that the cluster f_cl11 is correctly recognized as pristine ice with low Z and KDP and high ZDR and is identified during time periods when the height of the cloud top was law. Significant aggregation is unlikely in this situation due to low number concentration of ice particles and small difference in their terminal velocities.

    Thank you for the suggestion, we shall endeavour to explore the use of the vertical gradient of Z in future studies and consider how the clusters relate to cloud top height and/or temperature when considering their future microphysical interpretation.

7.  I notice that several literature references mentioned in a body of the manuscript (Kumjian 2012, Hampton 2019, 2020, Murphy 2018) are missing in the reference list.

    We apologize for missing this. It has now been added and the reference list has been checked for accuracy.

---

## Author Response (AR2)

[revised manuscript text omitted]

Reviewer 1:

**1) Page 3, Line 16-19: I suggest instead to the authors to underline as novelty the fact that QVP data are used. Previous studies already demonstrated the viability of the data-driven approach for hydrometeor classification and this is not a clear novelty of the paper (so i would rephrase also what stated at Page 18, Line 4-5).**

We appreciate the reviewer's concern about the novelty of the QVP based clustering, but in our view, it is not the only novelty. The clustering technique we use is hierarchical, and it is a subtle but important difference from previous data-driven approaches in the literature that needs to be highlighted as well. We have further underlined the novelty of using QVPs in p.3, L.16-19:

"The point of this study is not to create a set of cluster characteristics that could be applied to other datasets. Rather the goal is to demonstrate the viability of this type of data-driven methodology for creating a set of labelled clusters (i.e. hydrometeor classes). As such, the comparison to in situ data and labelling done as part of this study is only shown as an example of how this tool can be used."

"The point of this study is not to create a set of cluster characteristics that could be applied to other datasets. Rather the goal is to demonstrate the viability of this type of data-driven methodology for creating a set of labelled clusters (i.e. hydrometeor classes) based on QVP data. As such, the comparison to in situ data and cluster labelling within this study is only shown as an example of how this tool can be used. "

and it was already present in p.18, L. 4-5:

"This paper presents a new technique of hydrometeor classification from QVP. Note that both the hierarchical data-driven approach and the use of QVP is novel."

**2) Noise and measurement errors (Page 14/15, Lines 30-11): here I still see room to perform and show the result of actual tests about the effect that noise or biases (which can be artificially introduced) can have on the output of the clustering (number of clusters, size, statistics of the content).**

We agree that the impact of measurement errors is important. Still, because the clustering technique we utilise provides a hierarchical tree of clusters and the final number of clusters is not predefined, a series of tests looking at the impact of differing levels of artificially introduced noise or biases would be meaningless. Adding artificial noise to the data will most likely change the number of optimal clusters or even the structure of the hierarchical tree, but any comparison between the perturbed and original data results would not be possible as there will be no direct correspondence between two optimal sets of clusters. Furthermore, the sensitivity of our technique to the perturbation of the input data would only mean that the method performs as expected – changing the result based on the input properties. As such, this is why we think the pre-processing of the data and the reduction of statistical errors through the formation of QVPs is essential.

**Reviewer 2:**

**Comment 1. I would suggest specifying the factors being used to correct Z and Zdr for attenuation / differential attenuation using differential phase.**

We now include requested information in the text. We also changed the description of the attenuation technique. This change is due to a confusion in the versions of the pre-processed data used for the final clustering tests. We have now triple checked, and we can state that the data we utilise were corrected with the method provided in the description of the pre-processing.  We apologise for this confusion and are glad the reviewer asked us to check. The text of the manuscript is adapted p.7, LL.24-25:

"attenuation using the ZPHI method (Testud et al. 2000) where =0.27, b=0.78 and specific differential attenuation is 0.14 times the specific horizontal attenuation."

Testud et al. (2000)  has also been added to the list of references.

**Comment 3. The Griffin et al. (2018) paper is cited in the text but not included in the reference list.**

> We thank the reviewer for noticing it. The reference list has been updated accordingly.

**Comment 5. What kind of "technical limitations" prevented the authors to compute wet bulb temperature? It is hard to believe that vertical profiles of humidity were not available from the Met Office Unified Model for the cases examined.**

> The high-resolution reanalysis data we have access to from the UK Met Office does not provide a 3D wet-bulb field. The dataset's documentation indicates that there should be a field that contains the 'wet bulb freezing height ASL', however, an inspection of the data files reveals no field with that name, or anything relating to wet bulb temperature. Over the last six months, we have tried to contact someone with knowledge of the creation of these files to help with this issue. Unfortunately, that has so far been unsuccessful. There is a 3D relative humidity field; however, these are given only on a limited number of pressure levels, similar to the temperature profiles used. Therefore, interpolating moisture between pressure levels would not hold close to the same level of accuracy as interpolating temperature, and make estimating the wet-bulb freezing height in this way would not be reliable.

**Comment 7. The source of the Kumjian 2012 reference is not specified. Is this his ERAD 2012 conference paper of PhD thesis? I still do not see Hampton (2019, 2020) in the reference list. It would be better to refer to the full journal article of Murphy et al. (2020) in JTECH rather than to the Murphy's MS thesis.**

> We thank the reviewer for pointing out the missed specification in the references. We have corrected the reference in the text from "Kumjian (2012)" to "Kumjian et al. (2013)" as the first paper where QVP's as a median value per range were presented is that paper (p. 3 L. 24). We removed references Kumjian (2012) and Hampton (2019, 2020) from the reference list as they are not used in the text of the paper anymore. The reference to Murphy et al. (2020) paper is used in the text in place of Murphy's MS thesis (p.18, L.1) and the list of references has been updated.